# FAM111A protects replication forks from protein obstacles via its trypsin-like domain

Yusuke Kojima[1], Yuka Machida[1], Sowmiya Palani[2], Thomas R. Caulfield [3], Evette S. Radisky[4], Scott H. Kaufmann [1,5] & Yuichi J. Machida[1,5✉]

Persistent protein obstacles on genomic DNA, such as DNA-protein crosslinks (DPCs) and tight nucleoprotein complexes, can block replication forks. DPCs can be removed by the proteolytic activities of the metalloprotease SPRTN or the proteasome in a replication-coupled manner; however, additional proteolytic mechanisms may exist to cope with the diversity of protein obstacles. Here, we show that FAM111A, a PCNA-interacting protein, plays an important role in mitigating the effect of protein obstacles on replication forks. This function of FAM111A requires an intact trypsin-like protease domain, the PCNA interaction, and the DNA-binding domain that is necessary for protease activity in vivo. FAM111A, but not SPRTN, protects replication forks from stalling at poly(ADP-ribose) polymerase 1 (PARP1)-DNA complexes trapped by PARP inhibitors, thereby promoting cell survival after drug treatment. Altogether, our findings reveal a role of FAM111A in overcoming protein obstacles to replication forks, shedding light on cellular responses to anti-cancer therapies.

[1] Department of Oncology, Mayo Clinic, Rochester, MN 55905, USA. [2] Department of Biochemistry and Molecular Biology, Mayo Clinic, Rochester, MN 55905, USA. [3] Department of Neuroscience, Mayo Clinic, Jacksonville, FL 32224, USA. [4] Department of Cancer Biology, Mayo Clinic, Jacksonville, FL 32224, USA. [5] Department of Molecular Pharmacology and Experimental Therapeutics, Mayo Clinic, Rochester, MN 55905, USA. ✉email: machida.yuichi@mayo.edu

Collision of the DNA replication machinery with replication fork obstacles causes replication fork stalling and double-strand DNA breaks (DSBs), posing a great threat to genome integrity[1–3]. In particular, protein obstacles such as DNA-protein crosslinks (DPCs) have deleterious effects on replication forks due to their bulky nature. DPCs can form by various mechanisms, with differing sizes and DNA-crosslink structures[4–6]. For example, crosslinking chemicals, such as formaldehyde and reactive oxygen species, covalently link proteins to DNA non-specifically. In contrast, abortive enzymatic reactions can produce specific DPCs as represented by topoisomerase 1 cleavage complexes (TOP1ccs), which result from trapping of TOP1 covalently bound to DNA in an intermediate step of the enzyme reaction. Because of the severe consequences of blocking replication forks, induction of DPCs or nucleoprotein complexes is a common mechanism for anti-cancer drugs. TOP1 inhibitors, such as topotecan and its prototype camptothecin (CPT), poison TOP1, thereby inducing cytotoxic levels of TOP1ccs[7]. On the other hand, poly(ADP-ribose) polymerase (PARP) inhibitors (PARPis) induce nucleoprotein complexes by trapping PARP1 protein at DNA single-strand breaks (SSBs) during the SSB repair process[8–10]. Through strong DNA binding that mimics a DPC, PARP1-DNA complexes block DNA replication and cause DSBs[11–13]. Since blocked replication forks and DSBs require the homologous recombination (HR) pathway for repair, PARPis are particularly toxic to HR-deficient tumors with BRCA1/2 mutations[14,15].

The discovery of the metalloproteases, Wss1 in yeasts and SPRTN in higher eukaryotes, revealed a previously unrecognized contribution of proteolysis to DPC repair[16–25]. The SprT metalloprotease domain of SPRTN contains a $Zn^{2+}$-binding sub-domain that binds single-strand DNA (ssDNA), which activates SPRTN[26]. SPRTN localizes to replication forks[20,27] and degrades DPC proteins upon replication fork collisions[28,29]. SPRTN deficiency causes progeria and an increased incidence of liver cancer in humans and mice, highlighting the importance of DPC repair[27,30–32]. In addition to SPRTN, DPCs can also be resolved using the proteasome, which can degrade DPCs following replication-coupled polyubiquitination of DPC proteins[29]. Furthermore, replication-independent sumoylation of DPC proteins has also been reported[29,33] and ACRC/GCNA family SprT proteases are involved in DPC repair in germline cells[33]. On the other hand, it is unknown whether these DPC repair mechanisms play a role in preventing replication fork stalling at non-covalently linked protein–DNA complexes. Indeed, we previously reported that Sprtn hypomorphic mouse embryonic fibroblasts (MEFs) are not hypersensitive to PARPis, implying that SPRTN is unlikely to be involved in removal of PARP1-DNA complexes[27]. Therefore, it is currently unknown whether a cellular mechanism exists to remove trapped PARPs, but such a system would be expected to undermine the therapeutic effect of PARPis.

In this study, we demonstrate that the putative serine protease FAM111A (Family with sequence similarity 111 member A) protects replication forks from stalling at PARP1-DNA complexes and TOP1ccs. FAM111A was originally identified as a replication fork protein by nascent chromatin capture proteomics and shown to interact with PCNA through its PCNA-interacting peptide (PIP) box[34]. Although it was proposed that FAM111A plays a role in PCNA loading, its function at ongoing replication forks was unknown. The presence of a trypsin-like protease domain in FAM111A and its localization to replication forks prompted us to investigate whether FAM111A is involved in DPC removal during DNA replication. Our findings uncover a critical role of FAM111A in promoting DNA replication fork progression not only at DPCs, but also at PARP1-DNA nucleoprotein complexes.

## Results

### FAM111A KO cells are sensitive to PARP and TOP1 inhibitors.

To investigate whether FAM111A is important for mitigating the effect of protein obstacles to replication forks, we examined whether FAM111A depletion affects sensitivities to a PARPi and a TOP1 poison. FAM111A-knockout (KO) by CRISPR/Cas9 (Fig. 1a, Supplementary Fig. 1, and Supplementary Table 1) rendered cells hypersensitive to the PARPi niraparib, which exhibits strong PARP-trapping effect[9], and the TOP1 inhibitor CPT (Fig. 1b). Similar results were also obtained with additional FAM111A KO clones obtained using CRISPR/Cas9 targeting different sequences of the gene (Supplementary Figs. 1, 2a, b and Supplementary Table 1). Furthermore, FAM111A KO cells were sensitive to another PARPi with a strong PARP-trapping effect[10], talazoparib (Supplementary Fig. 2c). In addition, FAM111A KO cells showed mild sensitization to etoposide (a topoisomerase II inhibitor that covalently traps TOP2), 5-aza-dC (a DNMT inhibitor that traps DNMT isoenzymes on DNA), and formaldehyde (Supplementary Fig. 2d). On the other hand, FAM111A KO cells were not sensitized to ionizing radiation (IR) or cisplatin (a DNA crosslinker) (Fig. 1b and Supplementary Fig. 2d).

The hypersensitivity of FAM111A KO cells to niraparib was also associated with a marked increase in apoptotic cell death as shown by Annexin V staining (Fig. 1c, d). Activation of the HR pathway was intact in FAM111A KO cells when they were treated with PARPi as shown by RAD51 focus formation (Supplementary Fig. 2e, f), indicating that an HR defect does not account for the PARPi hypersensitivity of FAM111A KO cells. This conclusion is also supported by the fact that FAM111A KO cells were not sensitive to IR (Fig. 1b), which relies on an intact HR pathway for optimal repair. Taken together, these results suggest that FAM111A protects cells from several anti-cancer drugs, especially PARP and TOP1 inhibitors.

### FAM111A directly binds to ssDNA through its central region.

Because FAM111A localizes to replication forks[34], we hypothesized that FAM111A might have DNA-binding activity. We tested this possibility by electrophoretic mobility shift assays (EMSA) using purified full-length FAM111A protein (Supplementary Fig. 3a) and various forms of labeled DNA oligonucleotides, including ssDNA, double-strand (ds) DNA, Y fork, and dsY fork (Supplementary Fig. 3b). FAM111A displayed robust binding to ssDNA and Y-fork DNA, weaker binding to dsDNA, and undetectable binding to dsY fork DNA (Fig. 2a). These results suggest that FAM111A exhibits preferential binding to ssDNA, as ssDNA structures were common between the ssDNA and Y fork DNA oligos used in these experiments. Supershift assays using anti-FAM111A antibody verified that the mobility shift was due to FAM111A rather than possible contaminants in the protein preparation (Fig. 2b). Binding of FAM111A to ssDNA was also confirmed using fluorescence polarization assays, which showed an equilibrium dissociation constant ($K_d$) of ~60 nM for ssDNA binding (Fig. 2c).

We next sought to map the DNA-binding domain of FAM111A. EMSA using various fragments of FAM111A (Fig. 2d and Supplementary Fig. 3c) revealed that the central region (176–282) was sufficient to bind to ssDNA (Fig. 2e). On the other hand, the ssDNA-binding activity of the N-terminal FAM111A fragment (1–282) was diminished with varying degrees when a part of the central region was deleted (Supplementary Fig. 3d–f). These results suggested that the central region (176–282) is necessary and sufficient for the ssDNA binding of FAM111A.

To identify amino acid residues in the FAM111A central region that are crucial for ssDNA binding, we introduced point mutations in aromatic or positively charged amino acid residues

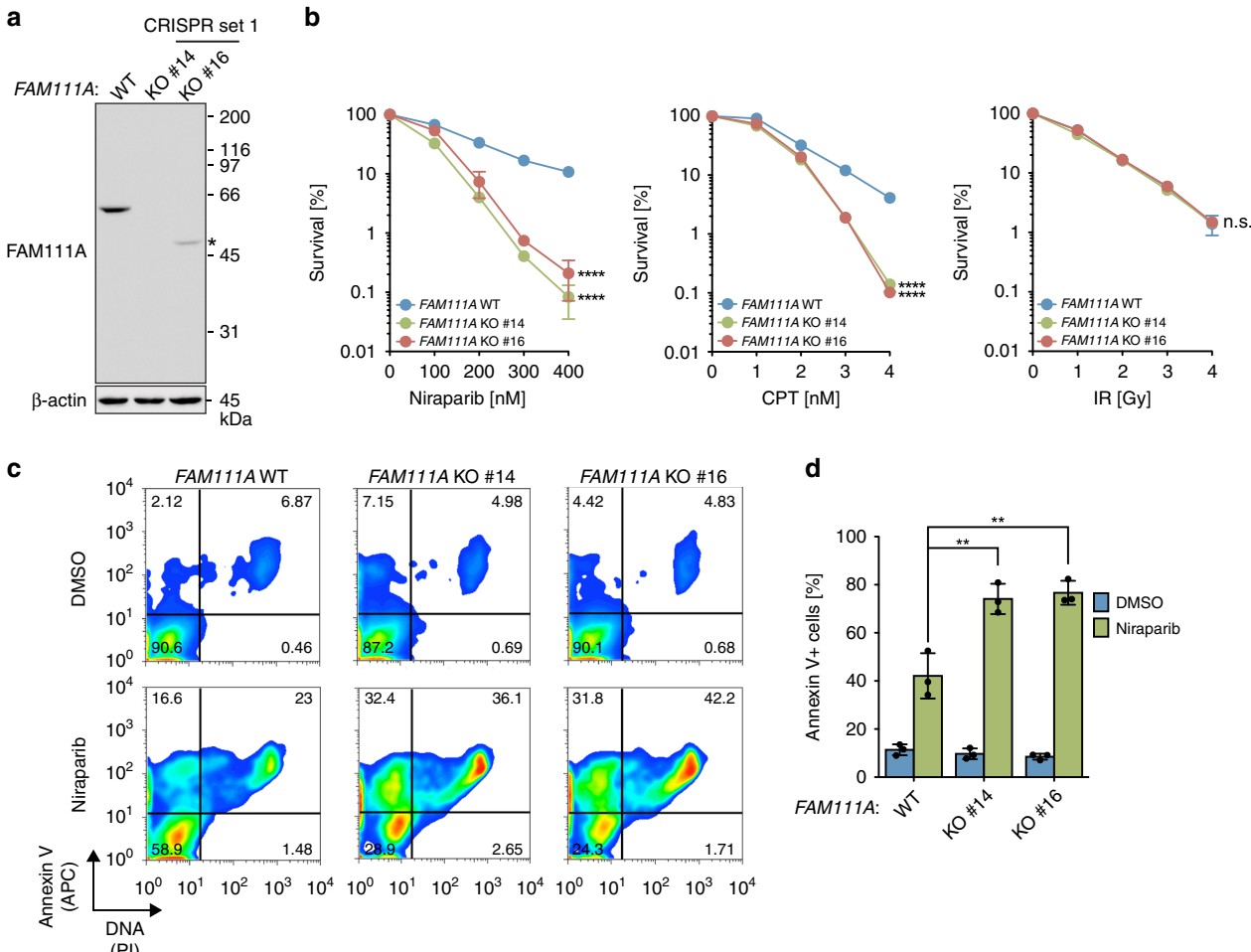

**Fig. 1 FAM111A deficiency sensitizes cells to PARPis and CPT. a** Knockout of *FAM111A* by CRISPR/Cas9 (set 1 gRNA). Parental HAP1 (WT) and *FAM111A* KO clones (KO #14 and #16) were analyzed for the indicated proteins by Western blotting. *, truncated FAM111A protein. **b** Clonogenic survival assays. Cells analyzed in **a** were cultured in the presence of the indicated concentration of niraparib or CPT for 6 days. For ionizing radiation (IR), cells were irradiated at the indicated doses and cultured for 6 days. Results shown are representative of three independent experiments and values are mean ± s.d. of technical replicates (*n* = 3). *p* values were calculated relative to *FAM111A* WT. ****$p < 0.0001$; n.s. not significant (two-tailed unpaired *t*-test). **c** Analyses of cell death by Annexin V/PI staining. After treatment with 1 μM niraparib or DMSO for 48 h, cells stained with Annexin V-APC and PI were analyzed by flow cytometry. Results shown are representative of three independent experiments and percentages of cells in each quadrant are indicated. **d** Quantification of Annexin V-positive cells. Experiments were performed as in **c**. Values are mean ± s.d. of independent experiments (*n* = 3). **$p < 0.01$ (two-tailed unpaired *t*-test). Source data are provided as a Source Data file.

based on a previous report showing that those amino acid residues are involved in ssDNA binding[35]. We particularly focused on amino acids conserved among species or stretches of positively charged amino acids (Supplementary Fig. 3g). EMSA using MBP-FAM111A 1–282 with point mutations revealed that the F231A mutation had the largest effect on ssDNA binding among the mutants tested (Supplementary Fig. 3h, i). The F231A mutation decreased DNA binding to both ssDNA and Y-fork substrates (Fig. 2f), and similar results were obtained with the minimum DNA-binding domain (176–282) (Supplementary Fig. 3j-l). All of these data suggest that Phe231 plays a key role in FAM111A binding to ssDNA.

**Disease-causing mutations enhance FAM111A cleavage in vivo.**
FAM111A contains a trypsin-like domain at its C-terminus with an intact catalytic triad consisting of His385, Aps439 and Ser541, a characteristic feature of the S1 serine protease family[36,37]. Given that proteases often exhibit autocleavage activity, we set out to find processed fragments of FAM111A in vivo to study the

putative protease activity. When exogenous FAM111A was expressed in 293T cells, which lack endogenous FAM111A expression for an unknown reason (Supplementary Fig. 4a), a small FAM111A fragment (~40 kDa), as well as full-length protein, was detected on the immunoblot using the anti-FAM111A antibody that recognizes the N-terminal region (Fig. 3a). This fragment was not observed with the mutant that contains a substitution of the active site serine to an alanine (S541A). These results raised the possibility that FAM111A might have autocleavage activity in vivo. Interestingly, germline mutations have been found in *FAM111A* in patients with Kenny–Caffey syndrome (KCS) and osteocraniostenosis (OCS)[38–41], and it has been postulated that these mutations may cause gain-of-function FAM111A activity because the mutations are always heterozygous and no loss-of-function mutations (i.e., deletions and truncations) have been reported[38]. We therefore tested whether the mutations found in KCS and OCS patients produce constitutively active FAM111A. For all the patient mutants tested (R569H and Y511H for KCS; S342del and D528G for OCS), the levels of full-length protein were reduced concomitant with a

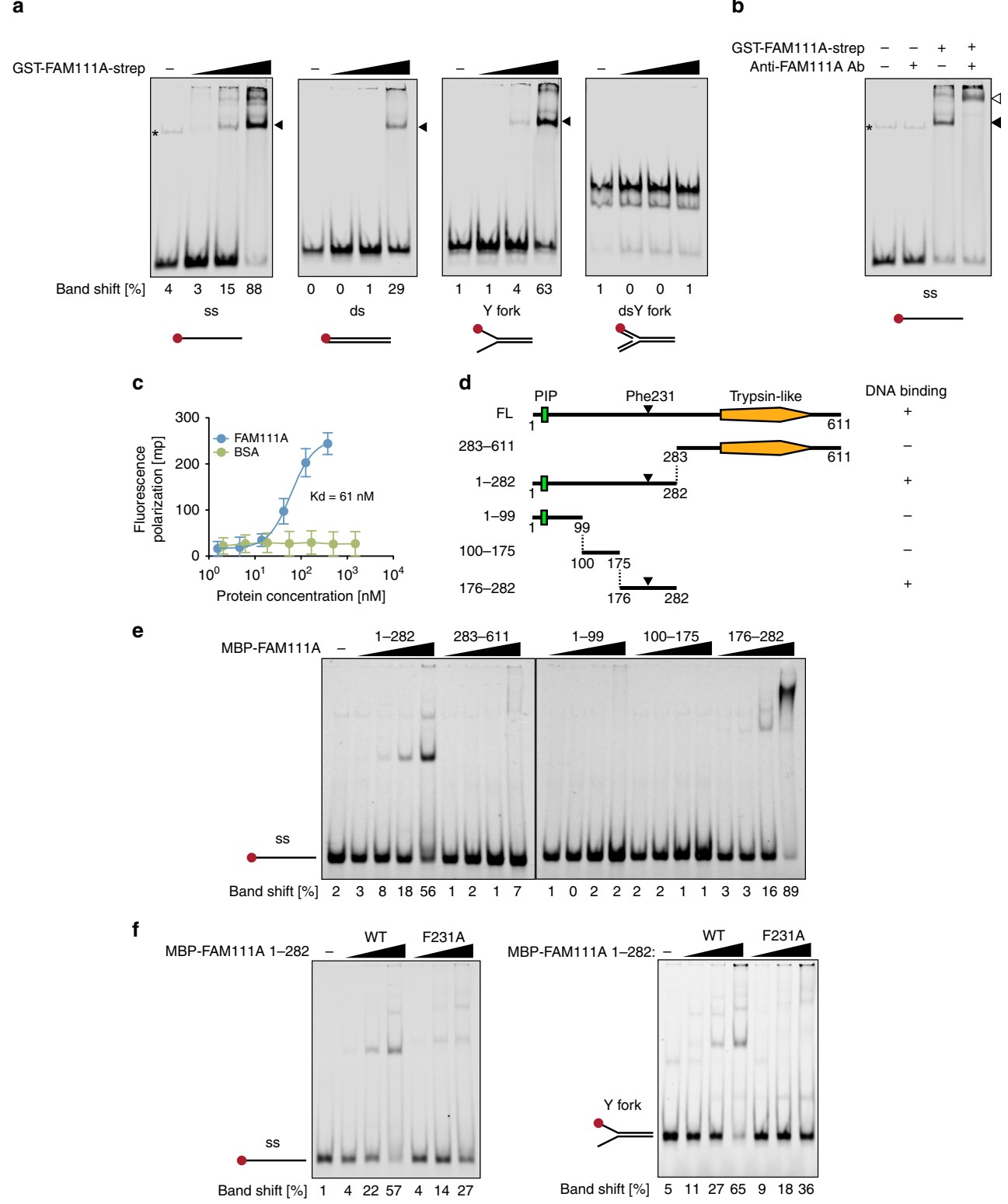

large increase in the levels of the processed fragment (Fig. 3b, lanes 4, 6, 8, and 10). Addition of an active site mutation (Ser541, S541A) to these patient mutants restored full-length protein expression and eliminated the processed fragment (Fig. 3b, lanes 5, 7, 9, and 11). These results suggest that the FAM111A patient mutants are constitutively active forms that result in the conversion of the full-length protein to smaller fragments through a possible autocleavage mechanism.

**FAM111A activity exhibits chymotrypsin-like specificity.** The S1 family serine proteases exhibit three major types of substrate specificities, which are determined by recognition of the P1 substrate residue (the amino acid residue preceding the cleavage site) by the S1 pocket of the enzyme. The trypsin-type prefers positively charged residues at the P1 position, whereas the chymotrypsin-type and the elastase-type prefer large and small hydrophobic residues, respectively[36,37]. To gain more information

**Fig. 2 FAM111A binds to ssDNA through its central region. a** Electrophoretic mobility shift assay (EMSA). DNA substrates labeled with IRDye700 (ss: single-stranded, ds: double-stranded, Y Y fork, dsY double-stranded Y fork) were incubated with increasing amounts of recombinant GST-FAM111A-strep protein (50, 150, and 450 nM). BSA (450 nM) was added in the "-" lanes. Positions of the shifted band are indicated by arrowheads. *, nonspecific band. **b** Supershift assay. GST-FAM111A-strep (450 nM) was pre-incubated with anti-FAM111A antibody on ice for 30 min before adding ssDNA oligos. Positions of the shifted and supershifted bands are indicated by a black and a white arrowhead, respectively. *, nonspecific band. **c** Fluorescence polarization (FP) DNA-binding assay. Various amounts of recombinant strep-FAM111A protein or BSA were mixed with 6-FAM-labeled ssDNA oligos and FP values were measured. Values are mean ± s.d. of independent experiments ($n = 3$). **d** Schematic representation of full-length (FL) and truncated FAM111A proteins used in this assay. Ability to bind to ssDNA oligos is summarized on right. PIP: PCNA-interacting peptide box. **e** Mapping the DNA-binding domain by EMSA using FAM111A fragments. IRDye700-labeled ssDNA oligos were incubated with increasing amounts of recombinant MBP-FAM111A proteins indicated on top (50, 150, 450, and 1350 nM). MBP (1350 nM) was added in the "-" lane. **f** Effect of the F231A mutation on ssDNA binding. Increasing amounts of MBP-FAM111A 1–282 proteins indicated on top (150, 450, 1350 nM for WT and equivalent amounts for F231A) were incubated with IRDye700-labeled ssDNA (left) or Y-fork (right) oligos. MBP (1350 nM) was added in the "-" lane. **a**, **b**, **e**, **f** Percentages of band shifts in each lane are shown below. Red dots indicate IRDye700. Source data are provided as a Source Data file.

on the substrate specificity of the in vivo FAM111A cleavage, we set out to determine the in vivo cleavage site of FAM111A. We expressed recombinant FAM111A proteins in insect cells and purified with the C-terminal Strep-tag. As was the case with FAM111A expressed in human cells, recombinant FAM111A produced in insect cells displayed a processed product that was more prominent with the FAM111A R569H mutant and absent when the active site was mutated (Fig. 3c). N-terminal peptide sequencing of the C-terminal fragment from the FAM111A R569H mutant identified the cleavage site to be C-terminal to Phe334 (P1) (Fig. 3d). Cleavage at this site is predicted to produce 38.4 kDa N-terminal and 31.5 kDa C-terminal fragments, consistent with the observed fragment sizes (Fig. 3a–c). Substitution of Phe334 with glycine or arginine diminished the in vivo cleavage of FAM111A (Fig. 3e), suggesting that Phe334 contributes to the specificity of the cleavage site. The hydrophobic residue (Phe334) at the P1 position implies chymotrypsin-type specificity for this cleavage.

Chymotrypsin-like proteases typically contain a hydrophobic S1 pocket[36,37]. In examining the structure model of the FAM111A trypsin-like domain based on the plant Deg1 protease[38,42], we indeed found a notable hydrophobic feature, Phe536, which is highly conserved among species, in the predicted S1 pocket (Supplementary Fig. 4b). In silico docking experiments using peptide sequences with the native cleavage site RTTFGKV as well as the mutant cleavage sites RTTRGKV and RTTGGKV (underlines indicate the P1 residue) showed the P1 Phe substrate displayed by far better docking score ($-12.9$ kcal mol$^{-1}$) compared with either the P1 Arg or P1 Gly substrates ($-10.5$ and $-8.5$ kcal mol$^{-1}$, respectively). Close examination of the S1 pocket revealed that the docked Phe peptide resulted in an edge-to-face ring stacking interaction between the P1 Phe334 and conserved Phe536 (Fig. 3f). By contrast, in the docking experiment with the P1 Arg mutant peptide, the Arg side chain was expelled from the S1 pocket during the energy minimization and relaxation protocol. These observations indicate that FAM111A is a serine protease with chymotrypsin-like specificity and strongly suggest that FAM111A exhibits autocleavage activity in vivo.

**FAM111A autocleavage can occur in *trans*.** Autocleavage can occur intramolecularly (in *cis*) and intermolecularly (in *trans*). Co-immunoprecipitation assays using FAM111A with different tags (3xFlag or EGFP) showed interaction between 3xFlag-FAM111A and EGFP-FAM111A (Fig. 3g), raising the possibility of an intermolecular mechanism. To directly assess this possibility, we co-expressed the constitutively active form of FAM111A (R569H) and its inactive counterpart (R569H/S541A) with different tags and tested whether constitutively active FAM111A causes cleavage of inactive FAM111A. Indeed, cleavage of the

inactive FAM111A occurred when the constitutively active FAM111A was co-expressed (3xFlag-FAM111A R569H/S541A with EGFP-FAM111A R569H and vice versa) (Fig. 3h, lanes 6 and 7), demonstrating that autocleavage of FAM111A can occur intermolecularly.

**FAM111A autocleavage requires an intact DNA-binding domain.** We next determined whether domains other than the trypsin-like domain are required for the FAM111A autocleavage. Introduction of the F231A mutation, which disrupts ssDNA binding of FAM111A (Fig. 2f, Supplementary Fig. 3j–l), but not YFAA, which diminishes PCNA interaction[34], prevented autocleavage of the constitutively active form of FAM111A (R569H) (Fig. 3i). Co-immunoprecipitation assays ruled out the possibility that the F231A and S541A mutations disrupted intermolecular interaction (Supplementary Fig. 4c). These data suggest that FAM111A activity in vivo is dependent on an intact ssDNA-binding domain.

**Trapping-dependent PARPi hypersensitivity after *FAM111A* KO.** In principle, PARPis can affect cell proliferation and survival by (i) diminishing formation of poly(ADP-ribose) polymer (inhibition mechanism) and (ii) blocking DNA replication through PARP1-DNA complex formation (trapping mechanism). The inhibition mechanism would be mimicked by *PARP1* KO, while the trapping mechanism would be eliminated by *PARP1* KO. To test which mechanism underlies the PARPi hypersensitivity conferred by *FAM111A* KO, we knocked out *PARP1* by CRISPR/Cas9 in the *FAM111A* WT and *FAM111A* KO backgrounds (Fig. 4a, Supplementary Fig. 1 and Supplementary Table 1). *FAM111A/PARP1* double KO (DKO) cells, as well as *PARP1* KO cells, were viable and indistinguishable from parental cells in proliferation assays (Fig. 4b). These data argue against PARP1 inhibition being the major mechanism of the hypersensitivity to PARPis in *FAM111A* KO cells. On the other hand, *PARP1* knockout conferred complete niraparib resistance to both *FAM111A* WT and *FAM111A* KO cells (Fig. 4c). Taken all together, these data suggest that sensitization to PARPis by *FAM111A* KO is dependent on PARP1 trapping.

**FAM111A deficiency causes TOP1cc accumulation.** The hypersensitivity of *FAM111A* KO cells to CPT (Fig. 1a, b) prompted us to hypothesize that FAM111A might also be involved in DPC repair in vivo. We tested this possibility by examining the levels of TOP1cc in *FAM111A* KO cells in the absence of CPT treatment. We found that *FAM111A* KO cells displayed increased TOP1cc foci compared with wild-type cells (Fig. 4d–f). Increased TOP1cc foci were also observed in U2OS cells after knockdown of FAM111A by RNAi (Supplementary Fig. 5a–c). TOP1cc accumulation in the *FAM111A* KO cells was

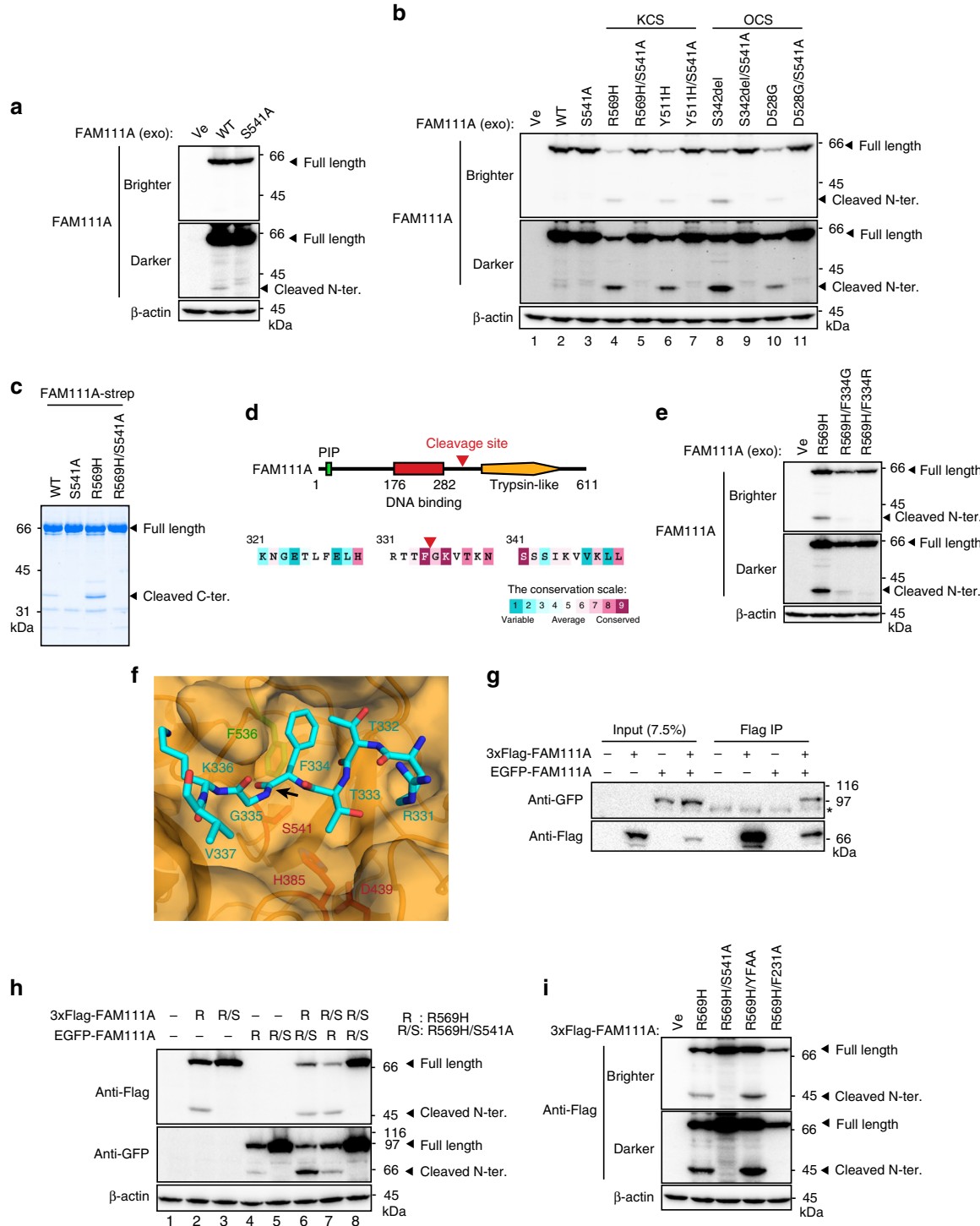

diminished by exogenous expression of wild-type FAM111A, but not the active site (S541A) or the PIP-box (YFAA) mutants (Fig. 4d–f), indicating the importance of these domains to prevent TOP1cc accumulation. These results indicate that FAM111A is important for the repair of TOP1ccs, providing an explanation for the hypersensitivity of *FAM111A* KO cells to CPT.

## FAM111A prevents fork stalling during PARPi or CPT treatment. 

Given that FAM111A localizes to replication forks[34], we assessed the importance of FAM111A in protecting replication forks from protein obstacles imposed by CPT or PARPis. For

these experiments, we employed DNA combing assays and examined the effect of the drugs on replication fork movement. Nascent DNA was sequentially labeled with CldU and then with IdU in the presence or absence of drugs (Fig. 5a). In the absence of drug treatment, replication fork movement was largely unaffected by *FAM111A* KO as seen in the comparable CldU tract lengths in *FAM111A* WT and KO cells (Fig. 5b). Although it has been reported that FAM111A may be involved in the PCNA loading[34], the amounts of PCNA on chromatin were largely unaffected in *FAM111A* KO cells (Supplementary Fig. 6a). In contrast, replication fork movement was severely affected by niraparib or CPT treatment in *FAM111A* KO cells as evidenced

**Fig. 3 In vivo cleavage of FAM111A. a** Detection of a cleaved FAM111A fragment. FAM111A proteins (WT or S541A) were transiently expressed in 293T cells and the indicated proteins were analyzed by Western blotting. Positions of the full-length and cleaved N-terminal fragment (N-ter.) are indicated. **b** Effect of mutations found in Kenny–Caffey syndrome (KCS) and osteocraniostenosis (OCS) on FAM111A cleavage. FAM111A proteins (WT or indicated mutants) were transiently expressed in 293T. The indicated proteins were analyzed by Western blotting. **c** Recombinant FAM111A-strep proteins. WT or indicated mutants of FAM111A-strep proteins were produced in insect cells and purified by the C-terminal strep tag. Positions of the full-length and cleaved C-terminal fragment (C-ter.) are indicated. **d** Identification of the FAM111A cleavage site by N-terminal Edman degradation. The position of the cleavage site is indicated in the schematic representation of FAM111A (upper) and in the amino acid sequence of FAM111A (lower). The color scale of the degree of amino acid conservation among species is shown below. **e** Effect of amino acid substitutions at the P1 site (Phe334) on in vivo cleavage of FAM111A R569H. The indicated FAM111A proteins were transiently expressed in 293T cells. Indicated proteins were detected by Western blotting. **f** Docked model of cleavage site residues (cyan) bound in the active site of FAM111A trypsin-like domain (orange). Catalytic triad residues are labeled in red and cleavage site is indicated by a black arrow. **g** Co-immunoprecipitation assays. 3xFlag-FAM111A and EGFP-FAM111A were transiently expressed in 293T as indicated and anti-Flag immunoprecipitation was performed. Input and precipitated proteins were analyzed by Western blotting. *, nonspecific band. **h** *Trans*-cleavage assays. 3xFlag-tagged and EGFP-tagged mutant FAM111A proteins (R: R569H, R/S: R569H/S541A) were transiently expressed in 293T cells. Expressed proteins were analyzed by Western blotting. **i** Effect of the mutations in the putative active site (S541A), PIP box (YFAA) and DNA-binding domain (F231A) on in vivo cleavage of FAM111A R569H. The indicated 3xFlag-FAM111A proteins were transiently expressed in 293T cells. Expressed proteins were analyzed by Western blotting. Empty vectors were used in "Ve" or "-" lanes.

by the reduced ratios of IdU- and CldU-labeled tract lengths (Fig. 5c, d). Exogenous expression of wild-type FAM111A, but not the S541A, or YFAA, rescued the effect of niraparib treatment in *FAM111A* KO cells (Fig. 5e), suggesting that the putative protease activity and PIP box of FAM111A are important to mitigate the effect of trapped PARPs. In addition, the F231A mutant did not rescue the replication fork defects during niraparib or CPT treatments (Fig. 5f, g), demonstrating the importance of the FAM111A DNA-binding domain. Importantly, the percentage of collapsed forks induced by hydroxyurea (HU) treatment was not increased in *FAM111A* KO cells (Supplementary Fig. 6b, c), suggesting that FAM111A does not regulate the stability of stalled replication forks. In addition, the S541A and F231A mutants of FAM111A localized properly to replication forks as assessed by iPOND assays (Supplementary Fig. 6d). These results demonstrate the importance of FAM111A in overcoming protein obstacles imposed by CPT and PARPis.

We reported previously that *Sprtn* hypomorphic MEFs are hypersensitive to CPT but not niraparib[27]. Together with the drug sensitivity data presented above (Fig. 1a, b), the previous observations raised the possibility that SPRTN might engage with TOP1cc but not PARP1-DNA complexes, while FAM111A might act on both. To investigate this possibility, we targeted *SPRTN* with CRISPR/Cas9 in HAP1, the same human cell line as *FAM111A* KO cells. This resulted in a *SPRTN* hypomorphic cell line (indicated as *SPRTN* CRISPR), which exhibited greatly reduced SPRTN levels due to KO in one allele and an eight-amino-acid deletion in the metalloprotease domain in the other allele (Fig. 5h, Supplementary Fig. 1 and Supplementary Table 1). Consistent with our previous study in mouse cells[27], the *SPRTN* CRISPR cells were sensitive to CPT, validating impairment of SPRTN function, yet they were not hypersensitive to niraparib (Supplementary Fig. 6e). DNA combing assays revealed that *SPRTN* CRISPR cells exhibited increased replication fork stalling with CPT but not with niraparib, whereas *FAM111A* KO displayed fork stalling with both drugs (Fig. 5i). These results demonstrate the unique function of FAM111A in processing PARP1-DNA complexes.

***FAM111A* KO causes enhanced DNA damage after PARPi treatment**. Finally, we investigated how *FAM111A* KO confers increased PARPi sensitivity and cell death (Fig. 1). If increased replication fork stalling (Fig. 5c) is the underlying cause, *FAM111A* KO cells are expected to experience cell-cycle defects and increased levels of DNA damage after PARPi treatment. Consistent with this prediction, *FAM111A* KO cells, which showed a modest reduction of G1 phase and increase in S and

G2/M phases in the absence of drug treatment, exhibited profound accumulation in G2/M phase upon treatment with niraparib (Fig. 6a, b). Moreover, *FAM111A* KO cells exhibited higher levels of phospho-Chk1, but not phospho-Chk2, than wild-type cells after niraparib treatment (Fig. 6c), indicating augmented activation of the ATR/Chk1 pathway, which typically responds to DNA replication stress. In addition, higher levels of DNA damage were observed in *FAM111A* KO cells than in wild-type cells after treatment with niraparib, as shown by immunostaining with the DNA damage marker γH2AX (Fig. 6d, e). Collectively, these results suggest that the absence of FAM111A causes replication fork stalling in S phase after PARPi treatment, leading to DNA damage, cell-cycle arrest in G2/M phases, and eventually cell death.

## Discussion
Advancing DNA replication forks face many obstacles due to proteins that are covalently crosslinked or tightly bound to DNA. In this study, we discovered that the replication fork protein FAM111A is important for resolving replication stalling caused by protein obstacles. We provide evidence for the protease activity of FAM111A in vivo, which is dependent on the DNA-binding domain identified in this study (Figs. 2 and 3). *FAM111A* KO cells exhibit TOP1cc accumulation and PARPi hypersensitivity that is dependent on PARP-trapping (Fig. 4). In the absence of FAM111A, protein obstacles induced by a TOP1 poison or a PARPi causes increased replication stalling (Fig. 5). The TOP1cc removal and fork protection by FAM111A require the PIP box and an intact trypsin-like domain (Figs. 4 and 5). Collectively, we propose a model that the FAM111A PIP box tethers the trypsin-like protease domain to replication forks to protect them from protein obstacles (Fig. 6f).

The role of FAM111A in resolving protein obstacles at replication forks is reminiscent of SPRTN's function in replication-coupled DPC repair. Both proteins contain a protease domain, a PIP box, and a DNA-binding domain that is necessary for protease activation. In addition, both proteins localize to replication forks and facilitate DNA replication at DPCs, including TOP1ccs induced by CPT. Despite these similarities, however, a striking difference was noted between FAM111A and SPRTN in the response to PARPis. Whereas CPT induces protein obstacles covalently crosslinked to DNA, PARPis are generally thought to trap PARPs non-covalently[9–13], although covalent PARP1-DNA complexes have also been reported[43,44]. Interestingly, it was recently demonstrated in *Xenopus* egg extracts that SPRTN activation requires polymerase collision with DPCs followed by helicase bypass of DPCs[29,45].

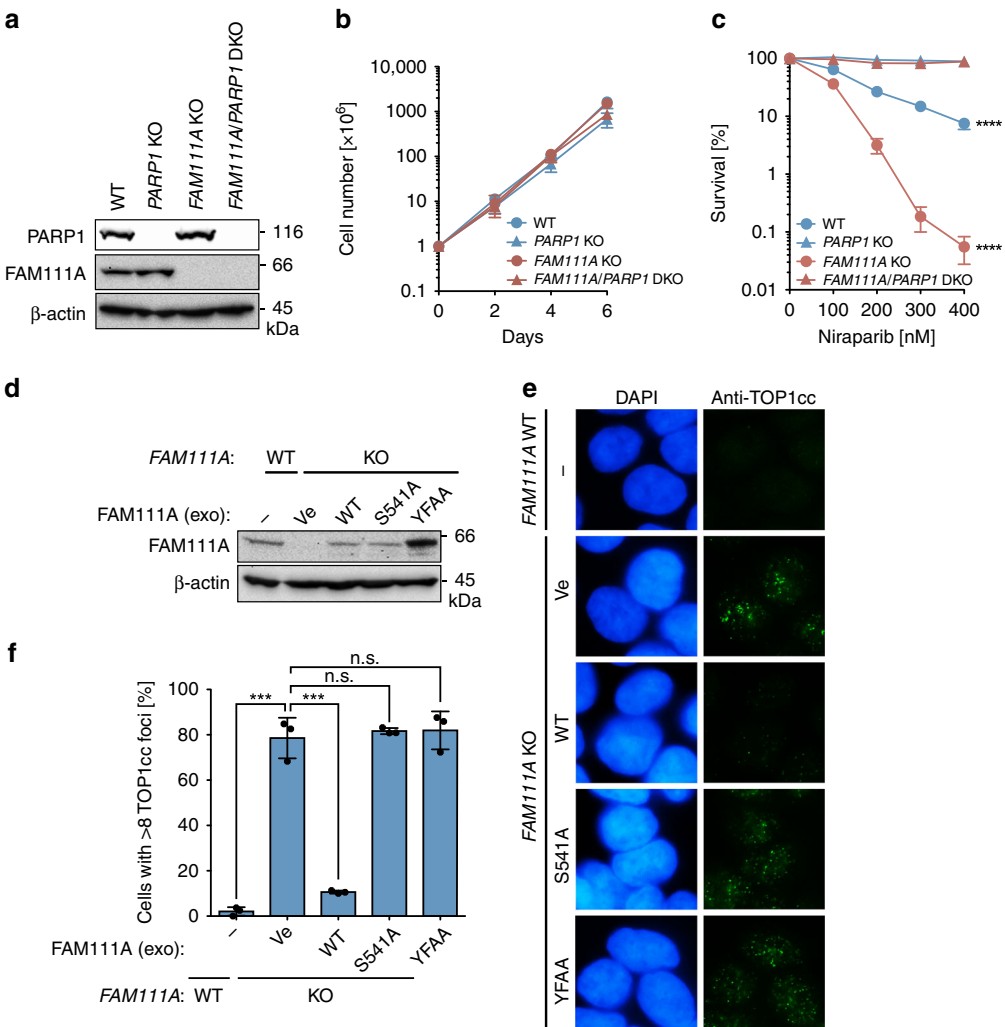

**Fig. 4 *FAM111A* KO causes trapping-dependent PARPi sensitization and TOP1cc accumulation. a** Knockout of *PARP1* by CRISPR/Cas9. Parental HAP1 (WT), *PARP1* KO #17, *FAM111A* KO #14, *FAM111A/PARP1* double knockout (DKO) #1 were analyzed for expression of the indicated proteins by Western blotting. **b** Cell proliferation assays. Cells analyzed in **a** were cultured and cell numbers were counted every other day up to 6 days. Values are mean ± s.d. of independent experiments ($n = 3$). **c** Clonogenic survival assays. Cells analyzed in **a** were cultured in the presence of the indicated concentrations of niraparib for 6 days. Results shown are representative of three independent experiments and values represent mean ± s.d. of technical replicates ($n = 3$). $p$ values were calculated between WT and *PARP1* KO and between *FAM111A* KO and *FAM111A/PARP1* DKO, respectively. ****$p < 0.0001$ (two-tailed unpaired $t$-test). **d** Exogenous (exo) expression of WT and mutant FAM111A proteins in *FAM111A* KO cells. The indicated FAM111A proteins were stably expressed by lentiviral vectors in *FAM111A* KO #14 and analyzed by Western blotting. Parental HAP1 (WT) is shown as a reference. Ve empty vector. **e** TOP1cc focus formation. Cells analyzed in **d** were stained with anti-TOP1cc antibody and DAPI. Scale bar, 5 μm. **f** Quantification of cells containing TOP1cc foci. Experiments were performed as in **e**. At least 100 cells were scored by an investigator blinded to sample identity and percentages of cells with 8 or more foci are shown. Values are mean ± s.d. of independent experiments ($n = 3$). ***$p < 0.001$; n.s. not significant (two-tailed unpaired $t$-test). Source data are provided as a Source Data file.

Therefore, the difference between SPRTN and FAM111A in the response to PARPis might be attributable to whether the fork-blocking proteins are covalently crosslinked to DNA or not. SPRTN might be specialized for protein obstacles covalently bound to DNA, while FAM111A might have a broader role in responding to protein obstacles regardless of the DNA-crosslinking status. Although further investigations are necessary to understand how different types of protein obstacles are cleared at replication forks, our study suggests that cells are equipped with multiple mechanisms to deal with various protein obstacles during DNA replication.

Our study strongly suggests that FAM111A functions as a protease in vivo. A mutation in the putative active site serine abolishes its function in TOP1cc repair and replication during niraparib treatment (Figs. 4d–f and 5e). Although wild-type FAM111A shows limited protease activity when expressed in

cells, the FAM111A mutants found in KCS or OCS patients are constitutively active and display increased autocleavage activity in vivo (Fig. 3b). We speculate that FAM111A might undergo activation only when necessary, and the KCS/OCS mutants might have bypassed the required activation mechanism. Homology-based structure modeling suggests that the FAM111A trypsin-like domain is similar to the Deg/HtrA family proteases[38], which contain one or two PDZ domains that play regulatory roles during protease activation in response to various cellular stresses, including high temperature, abnormal pH, or unfolded proteins[42,46,47]. Although FAM111A lacks a PDZ domain, we instead identified a DNA-binding domain in FAM111A that is necessary for FAM111A activity in vivo (Figs. 2 and 3i). This raises the possibility that FAM111A might undergo activation by sensing ssDNA structures as has been demonstrated for SPRTN. However, ssDNA-binding alone might not be a sufficient

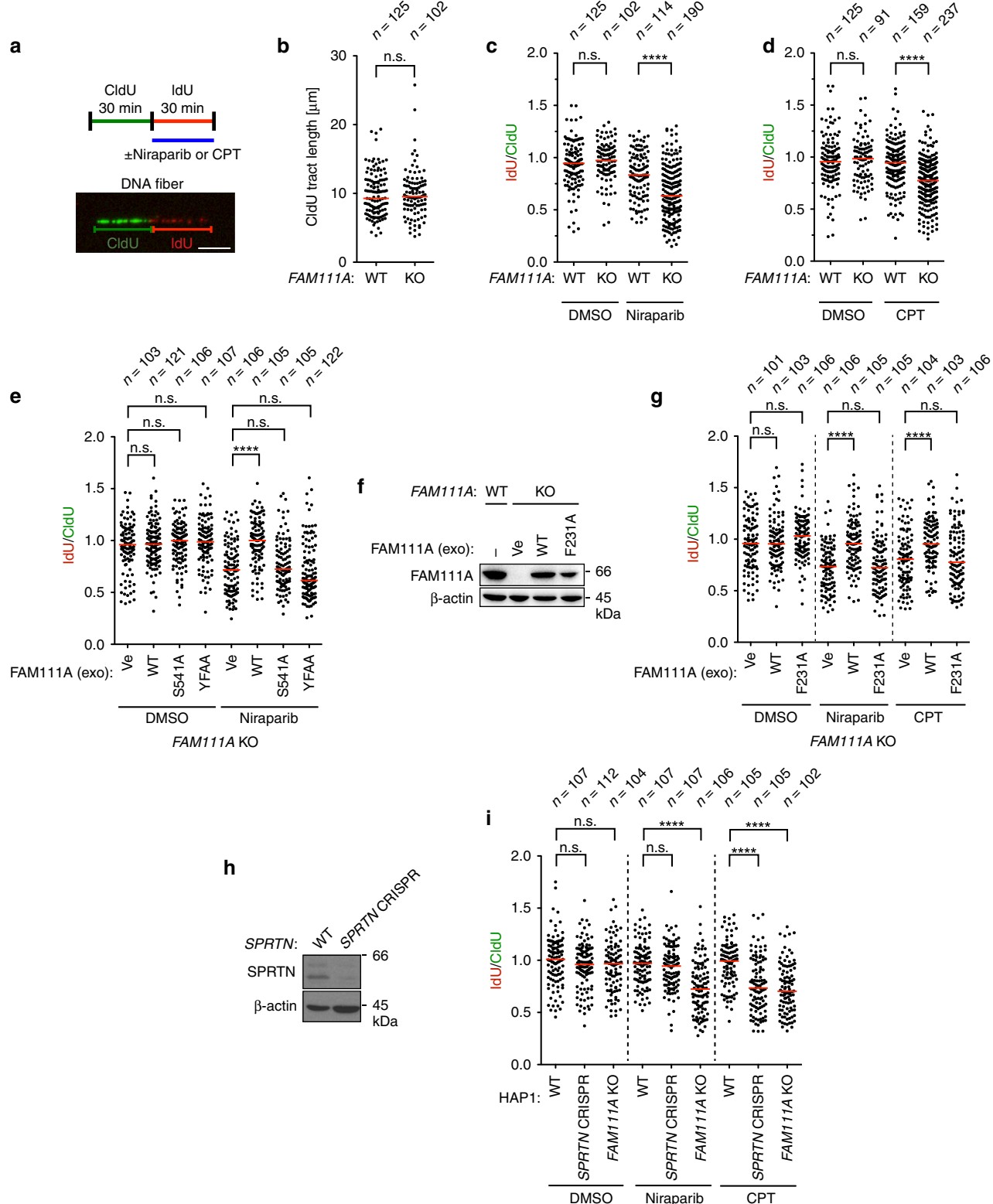

mechanism to restrict FAM111A activation to events of replication fork collisions with protein obstacles. Consistent with this notion, we have not been able to demonstrate protease activities in vitro using purified recombinant FAM111A even in the presence of multiple DNA structures. One possibility is that FAM111A might undergo additional regulation in vivo to become fully active, for example, through post-translational modifications, cofactor binding, or conformational changes. Interestingly,

many Deg/HtrA proteases dramatically change their oligomer status during activation, in some cases forming cages with the protease active sites facing inward[47,48]. Indeed, our results showed that FAM111A proteins interact with each other in vivo (Fig. 3g). While the activation mechanism is still unclear, our study suggests that FAM111A might be a stress-responsive protease similar to the Deg/HtrA family proteases, but one that responds to replication stress imposed by protein obstacles.

**Fig. 5 FAM111A is important to prevent replication fork stalling during PARP and TOP1 inhibitor treatment. a** Schematic representation of DNA combing assays. Nascent DNA was labeled with CldU (30 min) followed by IdU (30 min). Where indicated, niraparib or CPT was added during IdU labeling (upper panel). A representative picture of a replication track is shown (lower panel). Scale bar, 5 µm. **b** A dot plot showing the length of nascent DNA labeled by CldU in parental HAP1 (WT) and *FAM111A* KO #14. Distribution of the tract length is shown. **c, d** DNA combing assays. Parental HAP1 (WT) and *FAM111A* KO #14 were treated with 3 µM niraparib (**c**) or 30 nM CPT (**d**) during IdU labeling. Distribution of replication forks at different IdU/CldU ratios is shown. **e** DNA combing assays. *FAM111A* KO #14 expressing various FAM111A proteins (Fig. 4d) were treated with or without 3 µM niraparib during IdU labeling. Distribution of replication forks at different IdU/CldU ratios is shown. **f** Exogenous (exo) expression of WT and mutant FAM111A proteins in *FAM111A* KO cells. The indicated FAM111A proteins were stably expressed by lentiviral vectors in *FAM111A* KO #14 and analyzed by Western blotting. Parental HAP1 (WT) is shown as a reference. **g** DNA combing assays. *FAM111A* KO #14 cells expressing WT or the DNA-binding domain mutant (F231A) were treated with 3 µM niraparib or 30 nM CPT during IdU labeling. Distribution of replication forks at different IdU/CldU ratios is shown. **h** Depletion of SPRTN by CRISPR/Cas9 in HAP1 cells. The indicated proteins were analyzed by Western blotting. **i** DNA combing assays. Parental HAP1 (WT), *SPRTN* CRISPR #8 and *FAM111A* KO #14 were treated with 3 µM niraparib or 30 nM CPT during IdU labeling. Distribution of replication forks at different IdU/CldU ratios is shown. **b–e**, **g**, **i** Replication tracts were measured by an investigator blinded to sample identity. Horizontal red lines indicate median values and *p* values were obtained by two-tailed unpaired *t*-test (****$p < 0.0001$; n.s. not significant). *n* is the number of individual measures. Ve empty vector. Source data are provided as a Source Data file.

The present study also identifies FAM111A deficiency as a mechanism of hypersensitivity to PARPis. Because FAM111A deficiency augments DNA damage induced by PARPis (Fig. 6d, e), FAM111A inhibition might be a viable strategy to enhance PARPi killing in HR-deficient tumors and to expand PARPi therapies to HR-proficient tumors. Moreover, our data showed that FAM111A levels vary among cancer cell lines (Supplementary Fig. 4a). Therefore, determining the FAM111A status in tumors might be useful because a therapeutic advantage can be gained through the selective vulnerability of FAM111A-deficient cells to PARPis.

Finally, the present results shed important new light on previous studies of FAM111A. Our data suggest that FAM111A mutations found in KCS or OCS patients lead to constitutive protease activity (Fig. 3b). This observation raises two possibilities for the potential consequences of disease-associated mutations. One is that hyperactivation of FAM111A causes abnormal degradation of DNA-binding proteins in KCS and OCS, and the other is that hyperactivation causes a FAM111A deficiency through hyper-autocleavage. In fact, FAM111A forms oligomers and can degrade FAM111A proteins *in trans* (Fig. 3g, h), so that it is possible that the heterozygous mutations in KCS and OCS cause depletion of total FAM111A protein through a dominant-negative effect. Another implication of our study is that it might help understand the role of FAM111A as an antiviral defense mechanism. It has been demonstrated that FAM111A is a host range restriction factor for SV40 and mutant orthopoxviruses[49,50]. Given the recent report showing that FAM111A localizes to SV40 viral replication centers[51], it is possible that FAM111A might restrict viral replication by recognizing and degrading viral nucleoprotein complexes.

In conclusion, we demonstrated that FAM111A is important for DNA replication at protein obstacles. This study not only sets new frameworks for future studies on DPC repair, replication fork regulation, and anti-cancer therapies, but also provides new insight into the role of FAM111A in genetic diseases and viral defense mechanisms.

## Methods

**Cell culture.** Human chronic myelogenous leukemia cell line HAP1 was purchased from Horizon Discovery. Human embryonic kidney cell line 293T, human osteosarcoma cell line U2OS, human proximal tubular cell line HK2, human lung fibroblast cell line MRC-5, human cervical adenocarcinoma cell line HeLa, human colorectal carcinoma cell line HCT116, human mesothelioma cell line H226, human fibrosarcoma cell line HT1080, and human hepatocellular carcinoma cell line HepG2 were obtained from American Type Culture Collection. These cell lines were cultured in respective medium: Iscove's Modified Dulbecco's Medium for HAP1, Dulbecco's modified Eagle's medium for 293T and HeLa, McCoy's 5A medium for U2OS and HCT116, Eagle's minimum essential medium for MRC-5, HT1080, and HepG2. All of these media were supplemented with 10% fetal bovine serum (FBS). For HT1080 and HepG2, the medium was also supplemented with MEM non-essential amino acids (Corning) and sodium pyruvate (Corning). HK2 cell line was cultured in keratinocyte serum free medium (Invitrogen).

**Expression Plasmids.** For mammalian cell expression, a cDNA fragment encoding human FAM111A was cloned in the following vectors: pLVX2-IRES-puro (no epitope tag), pLVX3-IRES-puro (for an N-terminal 3xFlag tag), and pLVX6-IRES-puro (for an N-terminal EGFP tag). For insect cell expression, a DNA fragment encoding FAM111A with tandem Strep tags (FAM111A-strep) was synthesized with codon-optimization for expression in *S. frugiperda* (Thermo Fisher Scientific) (Supplementary Table 2). A DNA fragment encoding FAM111A with an N-terminal tandem Strep-tag (Strep-FAM111A) was generated by PCR. The ORFs were cloned into pDEST20 (for an N-terminal GST tag) and pFastBac NT-B. For bacterial expression, a DNA fragment encoding FAM111A 1–282 was generated by gene synthesis with codon-optimization for expression in *E. coli* (Thermo Fisher Scientific) (Supplementary Table 2) and used as a PCR template to amplify DNA fragments for FAM111A 1–99, 100–175, and 176–282. A DNA fragment encoding FAM111A 283–611 was amplified from human FAM111A cDNA. These PCR fragments were cloned in pMALII-c2x for expression of MBP fusion proteins in *E. coli*. Point mutations were introduced using the Quickchange mutagenesis kit (Agilent) or by PCR. All plasmids were confirmed by Sanger sequencing. Cloning primers are summarized in Supplementary Table 3.

**Transfection and viral infection.** Plasmid transfection was performed using Lipofectamine 2000 (Thermo Fisher Scientific) or Turbofectin 8.0 (OriGene) according to suppliers' instructions. Lentiviruses were produced by transfecting 293T cells with a lentiviral and two packaging plasmids (psPax2 and pMD2.G). Culture media containing lentiviral particles were harvested after 48 h and used to infect cells in the presence of 4 µg ml$^{-1}$ polybrane. After selection with 2 µg ml$^{-1}$ puromycin for 48 h, cells were cultured in the absence of puromycin for at least 24 h. siRNA transfection was performed using RNAiMAX (Thermo Fisher Scientific). The target sequences of siRNAs are as follows: siControl, 5′-CGUACGCG GAAUACUUCGA-3′; siFAM111A.271, 5′-CUAAAGAGCAACAGAAUAA-3′; siFAM111A.1213, 5′-CGAUUAAAGUAGUGAAACU-3′. siRNA oligo sequences are shown in Supplementary Table 3.

**Western blotting and co-immunoprecipitation.** Cells were lysed using NP-40 lysis buffer (50 mM Tris-HCl pH 7.4, 150 mM NaCl, 0.1% NP-40, 10% glycerol, 5 mM EDTA, 50 mM NaF, 1 mM Na$_3$VO$_4$) supplemented with protease inhibitor cocktail (Sigma). For FAM111A autocleavage assays, cells were lysed in SDS lysis buffer (PBS supplemented with 1% SDS and protease inhibitor cocktail). Cell lysates containing 30 µg protein were run on an SDS-PAGE gel and analyzed by Western blotting. Primary antibodies used were rabbit anti-FAM111A (Abcam, ab184572, 1:2000), rabbit anti-FAM111A (Sigma, HPA040176, 1:500), mouse anti-PARP1 (C-2-10, kind gift from Guy Poirier, 1:1000), mouse anti-SPRTN[52], mouse anti-β-actin (Sigma, A5316, 1:5000), rabbit anti-Chk1 (Epitomics, 1740–1, 1:2000), rabbit anti-phospho-Chk1 (S345) (Cell Signaling Technology, #2348, 1:1000), mouse anti-Chk2 (BD Bioscience, 611570, 1:1000), rabbit anti-phospho-Chk2 (T68) (Cell Signaling Technology, #2197, 1:1000), mouse anti-PCNA (Santa Cruz, sc-56, 1:2000), rabbit anti-Histone H3 (Cell Signaling Technology, #9715, 1:1000), mouse anti-Flag (Sigma, F1804, 1:1500), rabbit anti-Flag antibody (Cell Signaling Technology, #2368, 1:1000), and mouse anti-GFP (Santa Cruz, sc-9996, 1:1000). Chemiluminescence signals were imaged by X-ray film exposure or ChemiDoc (BioRad).

For co-immunoprecipitation experiments, 293T cells were transfected with expression plasmids for 48 h and lysed with NP-40 lysis buffer supplemented with proteinase inhibitor cocktail (Sigma) and 41 U ml$^{-1}$ Benzonase (Novagen). Cell lysates containing ~1 mg protein were incubated with anti-Flag beads (Sigma) for

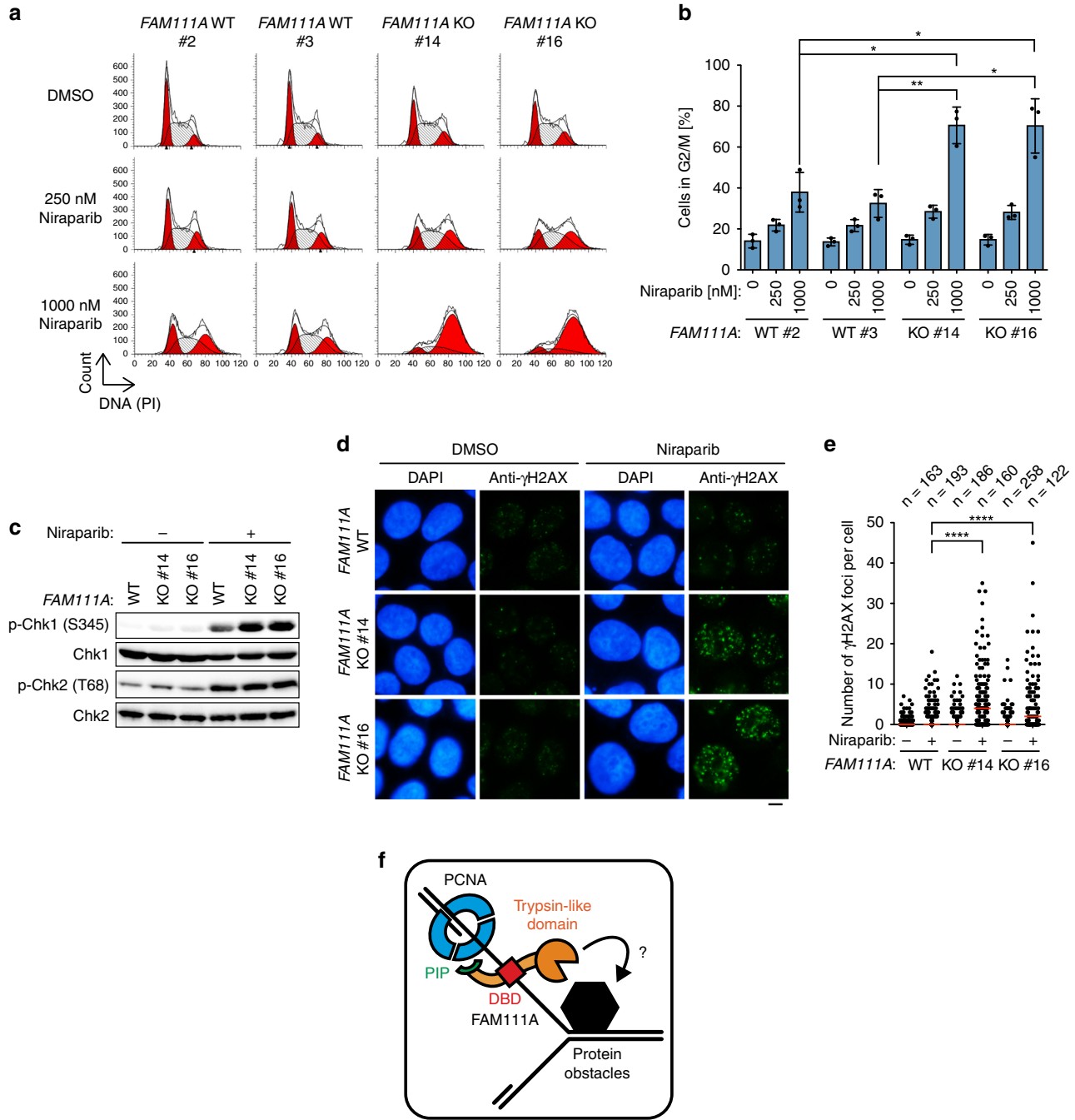

**Fig. 6 FAM111A deficiency causes replication stress, DNA damage and cell-cycle arrest during PARPi treatment. a** Cell-cycle profiling. Diploid HAP1 clones (WT #2 and #3) and *FAM111A* KO HAP1 clones (#14 and #16) were treated with indicated concentration of niraparib or DMSO for 24 h and analyzed by flow cytometry. Representative histograms from three independent experiments are shown. **b** Quantification of cells in G2/M phase. Experiments were performed as in **a**. Percentages of cells in G2/M phase were quantitated using ModFit. Values are mean ± s.d. of independent experiments ($n = 3$). *$p < 0.05$, **$p < 0.01$ (two-tailed unpaired *t*-test). **c** Levels of p-Chk1 and p-Chk2 after niraparib treatment. Indicated HAP1 cells were treated with 1.5 μM niraparib or DMSO for 24 h and indicated proteins were detected by Western blotting. **d** γH2AX foci after niraparib treatment. Parental HAP1 (WT) or *FAM111A* KO clones (#14 and #16) were treated with 250 nM niraparib or DMSO for 24 h and stained with anti-γH2AX antibody and DAPI. Scale bar, 5 μm. **e** Quantification of γH2AX foci per cell. Experiments were performed as in **d**. γH2AX foci were counted and the distributions of number of foci per cell are plotted (red line: median). ****$p < 0.0001$ (two-tailed unpaired *t*-test). *n* is the number of individual measures. **f** Schematic representation of the working model depicting the role of FAM111A at replication forks encountering protein obstacles. PIP PCNA-interacting peptide box, DBD DNA-binding domain. Source data are provided as a Source Data file.

4 h at 4 °C and the beads were washed five times with NP-40 lysis buffer. Precipitated proteins were analyzed by Western blotting.

**Generation of knockout cell lines**. CRISPR/Cas9-based gene knockout was performed using pX335 (kind gift from Feng Zhang, Addgene #42335) for *FAM111A*

and *PARP1*, and eSpCas9(1.1) (kind gift from Feng Zhang, Addgene #71814) for *SPRTN*. For CRISPR/Cas9 using the nickase Cas9 (pX335), the following pairs of gRNAs were used: *FAM111A*, set 1 (gRNA1 and gRNA2) and set 2 (gRNA3 and gRNA4); *PARP1*, gRNA1, and gRNA2. The target sequences of gRNA are as follows: *FAM111A* gRNA1, 5′-ATCAACCCACTAAGTGGCAC-3′; *FAM111A* gRNA2, 5′-CATATTATTGGCCATCCATAA-3′; *FAM111A* gRNA3, 5′-ATTATG

GTTGCCCACTTACT-3′; *FAM111A* gRNA4, 5′-CAATGTGTAAGGGTGACAT
T-3′; *PARP1* gRNA1, 5′-CCACCTCAACGTCAGGGTGC-3′; *PARP1* gRNA2,
5′-TGGGTTCTCTGAGCTTCGGT-3′; *SPRTN* gRNA5, 5′-AAGGGGTTCGCTG
AGACGGA-3′. Indel mutations were analyzed by Surveyor assays (Surveyor
Mutation Detection Kit, Integrated DNA Technologies) and Sanger sequencing of
PCR fragments cloned using TOPO TA Cloning (Thermo Fisher). Sequencing
results for cell clones used in this study are summarized in Supplementary Fig. 1
and Supplementary Table 1. All HAP1 clones were diploid based on FACS ana-
lyses. DNA oligos for cloning and PCR amplification of target regions are sum-
marized in Supplementary Table 3.

**Clonogenic survival assays.** Cells were seeded in triplicate in six-well plates,
allowed to attach on plates for 4 h, and treated with various agents. For CPT
(Sigma), niraparib (Merck), etoposide (Sigma), cisplatin (Teva Generics), and
5-aza-2′-deoxycytidine (Sigma), drug containing media were replenished every
other day. For talazoparib (Selleckchem), media were not changed during treat-
ment. For formaldehyde (Sigma), cells were treated for the first 24 h and the
medium was changed. Cells were cultured for 6 days, fixed, and stained with
Coomassie Blue. Colonies containing more than 50 cells were counted manually.

**Flow cytometry.** For detection of cell death, cells were stained first with Annexin
V-APC (BD Bioscience, 550474) in binding buffer (10 mM HEPES-NaOH pH 7.4,
140 mM NaCl, 2.5 mM CaCl$_2$) for 15 min at room temperature, and then with
100 μg ml$^{-1}$ of PI in binding buffer. For analyses of DNA content, cells were fixed
with 70% EtOH and stained with PI solution (50 μg ml$^{-1}$ PI, 10 μg ml$^{-1}$ RNase A,
0.05% NP-40). Two diploid clones of HAP1 (WT #2 and WT #3) were used as
controls for cell-cycle analyses. Stained cells were analyzed by BD FACS Canto II
(BD Biosiences). Percentages of cells in G2/M phase were estimated using ModFit
LT (Verity Software House).

**Immunofluorescence and microscopy.** For γH2AX and RAD51 immuno-
fluorescence, cells were fixed with 4% paraformaldehyde for 10 min, permeabilized
with 0.2% Triton X-100 in PBS for 10 min, and blocked with 1% normal goat
serum (for γH2AX) or 3% BSA (for RAD51) in PBS for 1 h. For TOP1cc staining,
cells were fixed with 4% paraformaldehyde for 15 min at 4 °C, permeabilized with
0.25% Triton X-100 in PBS for 15 min at 4 °C and treated with 1% SDS in PBS for
5 min at room temperature. Cells were washed five times with wash buffer (0.1%
Triton X-100, 0.1% BSA in PBS) and blocked with 10% milk in 10 mM Tris-HCl
pH 7.4 and 150 mM NaCl. Primary antibodies used were mouse anti-phospho-
Histone H2A.X (Ser139) (Millipore, 05-636, 1:500), rabbit anti-RAD51 (Calbio-
chem, PC130, 1:300), and mouse anti-TOP1cc antibody[53] (1:100). Secondary
antibodies used were goat anti-mouse IgG Alexa fluor 488 (Invitrogen, A11029,
1:2000) and goat anti-rabbit IgG Alexa fluor 488 (Invitrogen, A11034, 1:2000).
Images were captured using a Zeiss Axio Scope.A1 fluorescent microscope
equipped with a CCD camera (Jenoptik). Foci were scored using ImageJ on images
captured with a 63× objective (for γH2AX) or counted manually (for RAD51 and
TOP1cc). At least 100 cells were scored for focus formation by blinded observers.

**DNA combing assay.** Cells were labeled with 100 μM CldU (Sigma) for 30 min,
washed three times with pre-warmed PBS, and labeled with 100 μM IdU (Sigma)
for an additional 30 min. After labeling cells with IdU, cells were immediately
washed three times with ice-cold PBS to inhibit DNA replication. For studying
DNA replication under genotoxic stress, the IdU labeling was performed in the
presence of a drug (3 μM Niraparib or 30 nM CPT). To measure collapsed forks,
cells were treated with 5 mM HU for 30 min after CldU labeling (30 min) and
released in IdU-containing media for 60 min. Where indicated, 5 μM ATR
inhibitor (VE-821) was added during CldU labeling and HU treatment. DNA
fibers were extracted in agarose plugs with FiberPrep DNA Extraction kit
(Genomic Vision) and stretched onto silanized coverslips using the FiberComb
Molecular Combing System (Genomic Vision). Combed DNA was dehydrated in
an oven at 65 °C for 4 h and denatured with 0.5 M NaOH and 1 M NaCl for 8
min. Samples were then washed three times with PBS for 5 min each on a shaker,
dehydrated sequentially in 70, 90, and 100% ethanol for 5 min each and dried at
room temperature for 10 min. Samples were blocked with Block Aid (Invitrogen,
B10710) at 37 °C for 30 min and incubated with rat anti-CldU antibody (Abcam,
ab6326, 1:25) and anti-IdU antibody (BD Biosciences, 347580, 1:5) in Block Aid
at 37 °C for 1 hr. After washing with PBS-T (0.05% Tween-20), samples were
incubated with goat anti-rat IgG Alexa 488 (Invitrogen, A11006, 1:200) and goat
anti-mouse IgG Alexa 594 (Invitrogen, A11032, 1:200) in Block Aid at 37 °C for
30 min. After washing with PBS-T, samples were mounted with ProLong Gold
(Invitrogen). DNA fibers were photographed on a Zeiss Axio Scope.A1 fluor-
escent microscope and the length of CldU and IdU tracts were measured by
blinded observers using ImageJ.

**Purification of recombinant proteins.** Recombinant protein production in insect
cells was performed essentially as described before[19] with modifications. pDEST20/
FAM111A-strep, pFastBac NT-B/FAM111A-strep and pFastBac NT-B/Strep-
FAM111A were transformed into the DH10Bac *E. coli.* Purified bacmids were
transfected into Sf21 in using Cellfectin II (Thermo Fisher Scientific) as instructed

by the manufacturer. After incubation at 28 °C for 3 days, virus-containing
supernatants (P1 virus) were transferred to suspension culture in Sf-900 III SFM
(Thermo Fisher Scientific) and viruses were amplified for 3 days at 28 °C (P2 virus).
Viral titer was measured by baculoQUANT (Oxford Expression Technologies) as
instructed. If necessary, the P2 virus was further amplified (P3 virus). For protein
production, Sf21 cells in Sf-900 III SFM supplemented with glucose, yeastolate, and
lactalbumin were infected with virus stocks at M.O.I of 10–20 and cultured for
3 days at 28 °C. These cells were collected, washed once with PBS, frozen in liquid
N$_2$, and stored at −80 °C until protein purification. For protein purification, cell
pellets were thawed and resuspended in lysis buffer (50 mM HEPES-NaOH pH 7.5,
1 M NaCl, 1 mM MgCl$_2$, 10% glycerol, 1% NP-40) supplemented with 4 U ml$^{-1}$
Benzonase (Novagen). For GST-FAM111A-strep, 1 mM PMSF and cOmplete Mini
EDTA-free Protease Inhibitor Cocktail (Roche) were added to the lysis buffer. For
FAM111A-strep and Strep-FAM111A, 1 mM DTT, 1 mM EDTA and 2 μM Pep-
statin A were added to the lysis buffer. After 30-min incubation on ice, samples
were sonicated and cell debris was removed by centrifugation (20,000 × g, 4 °C,
30 min). Strep-Tactin superflow resin equilibrated with lysis buffer was mixed with
lysates by rotation at 4 °C for 2–3 h. Resins were washed twice with lysis buffer and
once with wash buffer (50 mM HEPES-NaOH pH 7.5, 250 mM NaCl, 10% gly-
cerol). For FAM111A-strep and Strep-FAM111A, 1 mM DTT was included in the
wash buffer. Proteins were eluted with elution buffer (wash buffer with 10 mM d-
Desthiobiotin), frozen in liquid N$_2$, and stored at −80 °C.
   For protein production in *E. coli*, pMALII-c2x vectors were transformed into
BL21 (DE3). For MBP-FAM111A WT 1–99, 100–175, 176–282 and 283–611, cells
were cultured until OD$_{600}$ = 0.5–1.2 and protein expression was induced with
0.1 mM IPTG at 18 °C overnight. Other MBP-FAM111A proteins were produced
at the basal level expression without IPTG at 37 °C overnight. Cells were collected,
washed with PBS, frozen in liquid N$_2$, and stored at −80 °C until protein
purification. For protein purification, frozen cells were thawed and resuspended in
PBS with 1 mM PMSF and 250 μg ml$^{-1}$ lysozyme. After 30 min incubation on ice,
cells were sonicated and cell debris was removed by centrifugation (20,000 × g, 4 °C,
30 min). Amylose resin (New England Biolab) equilibrated with PBS was mixed
with lysates at 4 °C for 2–3 h. Resins were washed with MBP wash buffer (50 mM
Tris-HCl pH 8.0, 500 mM NaCl, 10% glycerol, 0.1% NP-40) three times and
proteins were eluted with MBP elution buffer (50 mM Tris-HCl pH 8.0, 100 mM
NaCl, 10% glycerol, 10 mM maltose). Purified proteins were frozen in liquid N$_2$
and stored at −80 °C.

**Electrophoretic mobility shift assay (EMSA).** EMSA was performed essentially
as described before[54] with modification. 5′ IRDye700-labeled 75-nt DNA oligo
(O3063) was synthesized by IDT. Double-stranded (ds), Y-fork, and ds Y-fork
DNA substrates were generated by annealing labeled O3063 with other non-labeled
DNA oligos (ds: O4786; Y-fork: O4787; ds Y-fork: O4787 + O3730 + O3757) in
annealing buffer (10 mM Tris-HCl pH 8.0, 50 mM NaCl, 1 mM EDTA). DNA oligo
sequences are shown in Supplementary Table 3. DNA oligos were separated on 6%
non-denaturing polyacrylamide gels and the properly annealed DNA was extracted
in ddH$_2$O overnight at room temperature. Remaining gels were removed and the
buffer was exchanged to TE by Micro Bio-spin six column (Bio-Rad). For EMSA,
serially diluted recombinant FAM111A proteins were mixed with 0.7 pmol of DNA
substrates in binding buffer (final 25 mM Tris-HCl pH 7.5, 5 mM MgCl$_2$, 10%
glycerol, 20 ng μl$^{-1}$ BSA, 1 mM DTT, total 10 μl) and incubated on ice for 1 h. For
experiments using MBP-FAM111A 1–282, protein purity was assessed and protein
amounts that contain roughly equal levels of full-length proteins were used.
Samples were run on 6% non-denaturing polyacrylamide gel using 0.5 × TBE (4 °C,
100–120 V, 90–100 min) and imaged with Odyssey (LI-COR) or ChemiDoc MP
(Bio-Rad). Images were analyzed using Image Studio Lite (LI-COR) or Image Lab
(Bio-Rad) and percentages of band shifts were calculated based on the band
intensities of free and shifted DNA substrates.

**Fluorescence polarization DNA-binding assay.** DNA binding was measured by
fluorescence polarization assays essentially as described before[20] with modifica-
tions. Serially diluted FAM111A protein (Strep-FAM111A or MBP-FAM111A
176–282) or control protein (BSA or MBP) was mixed with 5′ 6-FAM-labeled 31-
nt DNA oligos (IDT) (final 10 nM) in FP buffer (10 mM HEPES-NaOH pH 7.5,
150 mM NaCl, total 30 μl). DNA oligo sequences are from previous study
(O1676)[54] (Supplementary Table 3). Samples were transferred to black 384-well
plate and fluorescence polarization was measured on a Cytation 1 imager (BioTek).
Kinetic constants were calculated by four-parameter logistic curve fitting in Prism 5
(GraphPad).

**Peptide sequencing.** Recombinant FAM111A-strep R569H protein was produced
in Sf21 cells and strep-tag containing protein fragments were captured by Strep-
Tactin Super Flow Plus as described above. The sample was separated by SDS-
PAGE and transferred to a PVDF membrane. Proteins were stained with Coo-
massie Brilliant Blue R-250 and the cleaved C-terminal fragment was sequenced by
Edman degradation peptide sequencing (five cycles) at Alphalyse Laboratories.

**Substrate docking analyses.** Homology modeling of the FAM111A trypsin-like
domain (residues 343–587) was performed using SWISS-MODEL[55]. The homology

ARTICLE

model was constructed using as a template the crystal structure of the plant protease Deg1 in active conformation (Protein Data Bank ID: 3QO6 chain A)[42], possessing 23.7% amino acid identity to the modeled region of FAM111A. The substrate peptide sequences were initially positioned in the active site by backbone alignment with corresponding residues of a peptide substrate present in the Deg1 crystal structure.

In preparation for docking, the structural models were imported using Protein-Preparation-Wizard with Maestro 2019 version 19–3 (Schrödinger, LLC) with the OPLS3 force field. Bond orders were assigned, zero-order bonds to metals were determined, disulfide bonds were created, and all hydrogens were generated. Hydrogen bond assignments were applied based on water sampling, and protonation states were predicted for pH 7.4 ± 1.0 using PROPKA[56,57]. Steric clashes were resolved using a convergence RMSD of 0.3 Å using the OPLS3 force field under Polak–Ribiere Conjugate Gradient (PRCG) minimization[58] for 5000 steps, or until the energy difference between subsequent structures was <0.001 kJ mol-Å$^{-156}$.

For molecular docking of peptide substrate sequences, substrates were docked into the binding site using Glide extra precision (XP) peptide-docking protocols (Glide, 19–3, Schrödinger, LLC), which allowed for multiple molecular conformations to be sampled[59–62]. The starting conformations based on the PRCG minimized state were further energy minimized with a water-based solvent to generate charges at an extended cutoff (VdW at 8.0 Å, electrostatic at 15 Å, H-bond at 4.0 Å). Soft restraints were placed on all residues >12 Å from the modeled substrate by using harmonic restraints of 100 kcal mol$^{-1}$, thus allowing the residues within the cutoff to freely refine position. In addition, the Monte Carlo Molecular Dynamics (MCMD) module with Maestro was employed for sampling side chain conformations and refining atomic positions of substrate-contacting residues within the FAM111A trypsin-like domain to optimize fitting with the substrate, as previously described[62–67]. Docking scores were used to rank the three sequences docked, namely, RTT**F**GKV, RTT**R**GKV, and RTT**G**GKV[60–62].

**Sequence analyses.** The degree of amino acid conservation among species was analyzed by the ConSurf server[68].

**Cell fractionation.** Cell fractionation was performed as described previously[9]. Briefly, trypsinized cells were washed with PBS and resuspended in hypotonic buffer (100 mM MES-NaOH pH 6.4, 1 mM EDTA, 0.5 mM MgCl$_2$) supplemented with protease inhibitor cocktail. After mixing by gentle pipetting, samples were layered on hypotonic buffer containing 30% sucrose and centrifuged for 10 min at 15,000 × $g$ at 4 °C. The pellet was isolated and proteins were extracted sequentially by resuspending the pellet in buffer A (50 mM HEPES-NaOH pH 7.5, 100 mM KCl, 2.5 mM MgCl$_2$, 0.05% Triton X-100, protease inhibitor cocktail), buffer B (50 mM HEPES-NaOH pH 7.5, 250 mM KCl, 2.5 mM MgCl$_2$, 0.05% Triton X-100, protease inhibitor cocktail) and buffer C (50 mM HEPES-NaOH pH 7.5, 500 mM KCl, 2.5 mM MgCl$_2$, 0.1% Triton X-100, protease inhibitor cocktail). In each step, pellet was isolated by centrifugation (15,000 × $g$, 4 °C, 10 min). After these extractions, chromatin proteins were released by micrococcal nuclease (NEB) in buffer A supplemented with 5 mM CaCl$_2$ at 37 °C for 15 min. Samples were centrifuged (15,000 × $g$, 4 °C, 10 min) and released proteins in supernatant were analyzed by Western blotting.

**iPOND.** 3xFlag-FAM111A (WT, S541A or F231A) were stably expressed in 293T cells by lentiviral vectors and iPOND experiments were performed essentially as described previously[69]. Briefly, cells were labeled with 10 μM EdU for 20 min, and chased with 10 μM thymidine for 30 min. Harvested cells were fixed with 1% formaldehyde for 20 min, permeabilized by 0.25% Triton X-100 for 30 min at room temperature followed by 90 min of click reaction, in which biotin is conjugated to EdU in the EdU-labeled DNA. Cells were lysed in RIPA lysis buffer (50 mM Tris-HCl pH 7.4, 150 mM NaCl, 1 mM EDTA, 1% NP-40, 0.1% SDS, 0.5% Sodium deoxycholate) supplemented with protease inhibitor cocktail and PMSF and sonicated. Precipitates were removed by centrifugation (15,000 × $g$, 15 min, 4 °C) and EdU-labeled DNA-protein complexes were pulled down by streptavidin agarose beads (EMD Millipore) at 4 °C overnight. Beads were washed once with RIPA lysis buffer, once with 1 M NaCl, once with PBS, and twice with RIPA lysis buffer. Pulled down proteins were eluted in 2xLithium dodecyl sulfate (LDS) sample buffer at 70 °C for 1 h and analyzed by Western blotting.

**Statistics and reproducibility.** GraphPad Prism 5 was used to graph the data and statistical significance was determined by a two-tailed unpaired $t$-test. Experiments were repeated at least twice and similar results were obtained.

**Reporting summary.** Further information on research design is available in the Nature Research Reporting Summary linked to this article.

## Data availability
The source data underlying Figs. 1b, d, 2a, c, e, f, 4b, c, f, 5b–e, g, i, 6b, e, and Supplementary Figs. 2b–d, f, 3f, i, k, l, 5c, 6c, and e are provided as a Source Data file. The uncropped blots underlying Figs. 1a, 2a, b, e, f, 3a–c, e, g–i, 4a, d, 5f, h, 6c, and

Supplementary Figs. 2a, 3a–c, e, f, h–k, 4a, c, 5a, 6a, and d are provided in Supplementary Fig. 7. All data are available from the authors upon reasonable request.

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

## Acknowledgements

We thank Larry M. Karnitz for critical reading of the manuscript, Kevin L. Peterson for technical assistance, and Karen S. Flatten for technical advice. This work was supported by the National Institutes of Health (R01 CA233700 to Y.J.M.) and Eagles Cancer Research Fund to Y.J.M.

## Author contributions

Conceptualization, Y.J.M., Y.K., and S.H.K.; Investigation, Y.K., Y.M., Y.J.M., and S.P.; Formal analysis, T.R.C. and E.S.R.; Writing–Original Draft, Y.K. and Y.J.M.; Writing–Review & Editing, Y.M.J., Y.K., S.H.K., T.R.C. and E.S.R.; Funding Acquisition, Y.J.M.; Resources, S.H.K.; Supervision, Y.J.M.

## Competing interests

S.H. Kaufmann is an inventor on U.S. patent No. 8,530,172 held by Mayo Clinic that deals with use of antibodies to protein–DNA complexes as diagnostic reagents to predict responsiveness to topoisomerase inhibitors. The remaining authors declare no competing interests.
