## [Peer Review File · Nature Communications]

Reviewers' comments:

Reviewer #1 (Remarks to the Author):

The repair mechanisms of DNA protein crosslinks (DPCs) has recently gained a lot of attention since the discovery of a specialised DPC protease (Wss1, SPRTN) that degrades DPCs during DNA replication. These original findings have opened new areas of investigation, focused in understanding how these toxic lesions are removed from the genome, and raised the possibility that additional DPC proteases may exist to cope with the wide variety of DPCs.

In this manuscript Kojima et al. identify a protein that was previously detected at ongoing replication forks, to be an essential part in the response to DNA replication roadblocks. FAM111A contains a trypsin-like protease domain, a PCNA interacting motif and a newly identified ssDNA binding region and these domains are all essential for promoting replication progression through replication fork barriers induced by PARP1i or TOP1 poison. They further provide convincing evidence that FAM111A acts as a protease in vivo which is demonstrated by its auto cleavage activity. They conclude that FAM111A is a protease that participates in the repair of DNA-protein roadblocks during DNA replication to ensure that the replisome progresses through these impediments. This is further substantiated by the observations that FAM111KO cells accumulate TOP1 DNA crosslinks.

The results provided by Kojima et al. are sound, well-presented, and interesting. I did not see any major flaws with the manuscript and the experiments are well controlled. The main weakness of the manuscript is that they could not detect FAM111A proteolytic activity in vitro but there are many reasons that could explain this (discussed in the discussion) and considering their thorough work showing that FAM111A acts as a protease in vivo I do not think that this should impede publication. The following comments should be addressed:

1. As referenced by the authors previous work indicates that in the absence of FAM111A there is reduced PCNA loading and impaired S phase progression. Here, FAM111A KO cells appear to replicate DNA just fine. These discrepancies need to be addressed or at least discussed. Have the authors looked at PCNA status on chromatin in their FAM111A KO cells?
2. It was not clear why 293T cells do not express FAM111A. It would be nice to clarify the status of FAM111A expression in different cell types and cancer. For example, ACRC (a recently described Sprt-protease) is germline/stem cell specific. It would be important to understand in what cellular context FAM111A might become critical.
3. The rescue of PARP1i sensitivity of FAM111KO cells by PARP1KO is very compelling (Figure 4C). The error bars of the triplicate experiments need to be added.

Reviewer #2 (Remarks to the Author):

This manuscript by Kojima et al. describes experiments on understanding the impact of accumulation of DNA-protein covalent crosslinks and tight complexes in replicating mammalian cells. This topic is of major biological interest, especially in the area of cancer biology as a number of chemotherapeutic agents are known to induce accumulation of DNA-protein covalent crosslinks and cellular hypersensitivity to the agents. Features of the proteolytic resolution of DNA-protein covalent crosslinks are addressed in the present manuscript.

The manuscript reports the identification and characterization of the FAM111A protease. The experimental design in each of the experiments provides for convincing interpretations; the authors are commended for this. In the latter portion of the Results section, a surprising specificity

of FAM111A against the PARP1 DPC vs another prominent protease was revealed, and this information is likely to lead to important work in the future. In addition, the hypothesis involving replication fork collision with DPCs was addressed and some of the results are consistent with the hypothesis.

In closing, the work described makes use of a fairly wide range of experimental. Yet, in each case, the methods are appropriately applied and interpreted, such that the overall interpretations are convincing.

Reviewer #3 (Remarks to the Author):

Proteins trapped onto DNA or crosslinked to DNA are frequent fork obstacles. How the DNA replication machinery deals with such obstacles remains poorly understood but involve proteolysis mediated by SPRTN in mammals and Wss1 in yeast. Here, the lab of Machida reports that FAM111A is a trypsin-like protease involved in cell resistance to drugs that induce DPCs and DNA-PARP complexes. FAM111A exhibits binding to ssDNA and autocleavage activity and specific mutations that affect those activities were identified and characterized in vitro. FAM111A prevents the accumulation of TOP1cc upon CPT treatment and this function requires its protease activity. FAM111A is necessary to promote replication fork progression upon PARP inhibition to prevent accumulation of DNA damages. This function requires its interaction with PCNA, its ssDNA binding activity and its protease activity.

Overall the data are of good quality and the work is very well conducted. The findings provide novel insights into our understanding of how cells deal with DPC and DNA-PARP complexes. However, some comments must be addressed to establish a more direct links between FAM111A and the resolution of forks blocked by DNA-PARP complexes.

Major points

1. The authors made the very interesting observation that the FRAM111A KO cells are highly sensitivity to PARP inhibitors but not SPRTN KO cells. They claim that FRAM111A is necessary to overcome PARP trapped onto DNA. These finding are important since it is unknown how trapped PARP are removed from DNA and this finding are relevant for the clinics. However there is no direct evidence for this. To strengthen their conclusion, it is important to demonstrate that PARP-DNA complexes accumulate in the absence of FAM111A (as shown by the lab of Y. Pommier in Murai et al. cancer research 2012). Of note, the authors should strength in the main text that Niraparib and Talazoparib are more efficient in PARP trapping than olaparib.
2. It is surprising that the authors did not investigate further the cellular phenotypes of the disease-associated mutations of FAM111A. Does the R569A mutant, that show high level of autocleavage, impacts the cell sensitivity to CPT and PARP inhibitors, replication fork progression upon niraparib treatment (Fig 5), accumulation of DNA damages (Fig 6), and accumulation of TOP1cc (fig4). These data would broaden the impact of the paper.
3. Figure 2 and supplementary figure 3: the characterization of the mutants that do not bind ssDNA are not convincing enough. First, some truncations (176-208, Fig S3f and 176-282, Fig2e) exhibit high molecular weight DNA intermediates. What are those structures? Second, the F231A and K227A-R230A mutants are supposed to have a reduced binding to ssDNA according to the blots on Fig 2f and Fig S3i. However, faint intermediates are detected. It is important that the authors provide a quantification of these blots to estimate the reproducibility of these data, as well as a better estimation of the reduced binding to ssDNA. Also, do those mutants (F231A and K227A-R230A) exhibit a reduced binding to fork structures ?
4. Recombinant FAM111A exhibits intramolecular autocleavage in vitro (Fig3c). It would be informative to test if this activity is stimulated by ssDNA in vitro.

5. Figure 3H: The authors propose that intermolecular cleavage can occur in vivo, possibly mediated by an intermolecular interaction. It is important to check that the R569H S541A mutant exhibit no defect in intermolecular interaction. Similar question can be raised for the ssDNA binding mutant (F231A).

6. Figure 5: the data indicate that the catalytic site (S541A) and the ssDNA binding activity (F231A) are necessary to promote replication fork progression upon niraparib treatment. Are those mutants properly associated to replication forks or nascent strands ?

7. Figure 6c-e: the data indicate that DNA Damages accumulate in the absence of FAM111A upon niraparib treatment. It would be informative to have a better characterization of the replication stress response induced in response to unresolved PARP-DNA complexes. Which pathway is activated: ATR-Chk1 and/or ATM-Chk2 ? Does ssDNA accumulate at replication forks ?

Minor points:

Statistics are missing on Fig 4c, 4f, S2b, S2c, S2d, S2f, S5c

Fig 4c: stars on the legend and triangles on the survival curves. To be clarified.

Point-by-point response to the reviewers' comments and suggestions:

Reviewer #1:

“1. As referenced by the authors previous work indicates that in the absence of FAM111A there is reduced PCNA loading and impaired S phase progression. Here, FAM111A KO cells appear to replicate DNA just fine. These discrepancies need to be addressed or at least discussed. Have the authors looked at PCNA status on chromatin in their FAM111A KO cells?”

- Our cell fractionation experiments showed that the levels of chromatin-loaded PCNA were unaffected in *FAM111A* KO cells, suggesting that FAM111A is not always necessary for PCNA-loading. The results of the fractionation data are included in **Supplementary Fig. 6a** in the revised manuscript.

“2. It was not clear why 293T cells do not express FAM111A. It would be nice to clarify the status of FAM111A expression in different cell types and cancer. For example, ACRC (a recently described Sprt-protease) is germline/stem cell specific. It would be important to understand in what cellular context FAM111A might become critical.”

- As suggested by the reviewer, we assessed FAM111A protein levels in 7 additional cell lines (**Supplementary Fig. 4a**). The results suggest that FAM111A is expressed in both normal and cancer cell lines with the exception of 293T, although the expression levels varied among cell lines. While it is not clear why FAM111A is not expressed in 293T, it appears that FAM111A is widely expressed among various cell types. This is also supported by the Human Protein Atlas database, which shows relatively uniform expression of FAM111A in normal organs [<https://www.proteinatlas.org/ENSG00000166801-FAM111A/tissue>].

“3. The rescue of PARP1i sensitivity of FAM111KO cells by PARP1KO is very compelling (Figure 4C). The error bars of the triplicate experiments need to be added.”

- Error bars have been added to the graph in the revised manuscript.

Reviewer #2:

“the work described makes use of a fairly wide range of experimental. Yet, in each case, the methods are appropriately applied and interpreted, such that the overall interpretations are convincing.”

- We thank the reviewer for the positive comments.

Reviewer #3:

Major points

“1. The authors made the very interesting observation that the FAM111A KO cells are highly sensitivity to PARP inhibitors but not SPRTN KO cells. They claim that FAM111A is necessary to overcome PARP trapped onto DNA. These finding are important since it is unknown how trapped PARP are removed from DNA and this finding are relevant for the clinics. However there is no direct evidence for this. To strengthen their conclusion, it is important to demonstrate that PARP-DNA complexes accumulate in the absence of FAM111A (as shown by the lab of Y. Pommier in Murai et al. cancer research 2012).”

- We are pleased that the reviewer found our study “interesting” and “important.” We agree with the reviewer that direct evidence for the removal of trapped PARPs by FAM111A would strengthen our conclusion. However, current methodology is not up to this task. Our data suggest that FAM111A, which is recruited to stalled replication forks, removes PARP1-DNA complexes at these forks. The method of Murai et al. measures total levels of PARPs trapped throughout the nucleus by cell fractionation. Because the fork-blocking PARPs at replication forks would be only a tiny fraction of the total trapped PARPs, the method of Murai et al. will not specifically assess the PARP-DNA complexes of interest.
- Furthermore, the method requires co-treatment with MMS to increase trapped PARPs to a detectable level. In this extreme condition (PARPi + MMS), in which new PARP-trapping keeps happening, PARP1 removal from chromatin by FAM111A might be slower than on-going PARP trapping.
- Analyzing removal kinetics of trapped PARPs after PARPi wash-off would not be possible either, because trapped PARPs dissociate from DNA upon wash-off within 30 min [*Cancer Res.* 72(21): 5588-5599 (2012)].
- For these reasons, measuring accumulation of trapped PARPs caused by *FAM111A* KO is not feasible. Nonetheless, we have demonstrated increased TOP1cc levels in *FAM111A* KO cells (**Fig. 4d-f**), which was possible because of the adduct-specific antibody. This result provides an example of the accumulation of proteins that are trapped on DNA in *FAM111A* KO cells. In addition, we have shown that the protease active site of FAM111A is required for the resolution of PARPi-induced fork stalling (**Fig. 5e**) and that PARPi hypersensitivity of *FAM111A* KO cells is dependent on PARP-trapping (**Fig. 4c**). Collectively, these data strongly support our model that FAM111A removes trapped PARPs at replication forks.
- We modified the Discussion (first full paragraph on p. 14) to clarify how we built our model based on our data.

“Of note, the authors should strength in the main text that Niraparib and Talazoparib are more efficient in PARP trapping than olaparib.”

- We have included descriptions of PARP-trapping efficiencies of niraparib and talazoparib on p. 5 of the revised manuscript.

“2. It is surprising that the authors did not investigate further the cellular phenotypes of the

disease-associated mutations of FAM111A. Does the R569A mutant, that show high level of autocleavage, impacts the cell sensitivity to CPT and PARP inhibitors, replication fork progression upon niraparib treatment (Fig 5), accumulation of DNA damages (Fig 6), and accumulation of TOP1cc (fig4). These data would broaden the impact of the paper.”

- We agree with the reviewer that it would be important to understand the effect of disease-associated FAM111A mutations on DPC repair. However, our effort to study these mutations further was hampered by the difficulty to express disease-associated mutants at the same levels as that of wild-type FAM111A, presumably due to autocleavage and subsequent degradation of the mutants (see **Fig. 3b**).
- However, our study raised a possibility that disease-associated FAM111A mutants might exhibit adverse effects through non-specific degradation of DNA-binding proteins through hyperactivation (discussed in the Discussion). On the other hand, the instability of disease-associated FAM111A mutants raises another interesting possibility that these mutations might indirectly cause FAM111A deficiency through hyperactivation. In fact, we presented data that FAM111A forms oligomers and can degrade FAM111A proteins *in trans* (**Fig. 3g,h**) so that it is possible that the disease-associated mutations, which are usually heterozygous, cause depletion of total FAM111A protein through a dominant-negative effect. Such a notion would broaden the impact of our study, and it is now discussed on pp. 16-17 of the revised manuscript. We thank the reviewer for the helpful comment.

“3. Figure 2 and supplementary figure 3: the characterization of the mutants that do not bind ssDNA are not convincing enough. First, some truncations (176-208, Fig S3f and 176-282, Fig2e) exhibit high molecular weight DNA intermediates. What are those structures?”

- We reevaluated the effect of mutations (F231A and K227A/R230A) on DNA binding by introducing the mutations in the minimum DBD (176-282). In the case of F231A, results of EMSA were consistent with those obtained previously using larger fragment (1-282) (**Fig. 2f** and **Supplementary Fig. 3j,k**). Furthermore, FP DNA binding assays using these fragments also demonstrated that the F231A mutation reduced ssDNA binding (**Supplementary Fig. 3l**). On the other hand, we noticed that the K227A/R230A mutation dramatically changed physical character of the protein (the solubility of DBD was significantly reduced). Because this raised a concern that the K227A/R230A mutation might affect overall structure of the protein, the K227A/R230A mutant was not pursued in the revised manuscript.
- The highly shifted band in 176-282 in **Fig. 2e** is likely caused by binding of multiple proteins on a single DNA oligo, because the supershifted bands start to appear progressively when protein concentrations were increased (this is more apparent in the new data in **Supplementary Fig. 3k**). We do not know why Δ 176-208 caused a greater shift than other fragments in **Fig. S3f**, but it is known that the degree of mobility shift in EMSA does not always correlate with the size of the protein [*Nat Protoc* 2(8): 1849-61 (2007)].

“Second, the F231A and K227A-R230A mutants are supposed to have a reduced binding to ssDNA according to the blots on Fig 2f and Fig S3i. However, faint intermediates are detected. It is important that the authors provide a quantification of these blots to estimate the reproducibility of these data, as well as a better estimation of the reduced binding to ssDNA.”

- As suggested by the reviewer, we quantified DNA binding based on band intensities of free DNA substrates and shifted substrates. These values were indicated below the figures of EMSA as percentages of band shifts. Reproducibility of the quantification has been confirmed by the repeated experiments shown in the Source Data file.

“Also, do those mutants (F231A and K227A-R230A) exhibit a reduced binding to fork structures ?”

- EMSA experiments showed that MBP-FAM111A 1-282 harboring the F231A mutation exhibited reduced binding to Y-fork structures (**Fig. 2f** in the revised manuscript). Similar results were also obtained with the minimum DBD (176-282) as shown in **Supplementary Fig. 3k** in the revised manuscript. The text at the top of p. 7 was modified accordingly.

“4. Recombinant FAM111A exhibits intramolecular autocleavage in vitro (Fig3c). It would be informative to test if this activity is stimulated by ssDNA in vitro.”

- The SDS-PAGE gel in **Fig. 3c** shows recombinant proteins after purification without *in vitro* reactions, so the small C-terminal fragment is a product of autocleavage that occurred *in vivo*. These purified recombinant proteins did not display autocleavage activity even when the reactions were supplemented with ssDNA. Possible explanations for this are discussed in the Discussion section.

“5. Figure 3H: The authors propose that intermolecular cleavage can occur in vivo, possibly mediated by an intermolecular interaction. It is important to check that the R569H S541A mutant exhibit no defect in intermolecular interaction. Similar question can be raised for the ssDNA binding mutant (F231A).”

- Co-IP experiments demonstrated that the S541A and F231A mutations did not disrupt intermolecular interaction. The results were included in **Supplementary Fig. 4c** and the manuscript was modified accordingly on p.10.

“6. Figure 5: the data indicate that the catalytic site (S541A) and the ssDNA binding activity (F231A) are necessary to promote replication fork progression upon niraparib treatment. Are those mutants properly associated to replication forks or nascent strands ?”

- We examined whether the S541A or F231A mutation affects the association of FAM111A with replication forks using iPOND assays [*Genes & Dev.* 25:1320-1327 (2011)]. The results showed that the mutant FAM111A proteins retained proper localization to nascent DNA. The results were included in **Supplementary Fig. 6d** and the manuscript was modified accordingly on p. 12.

“7. Figure 6c-e: the data indicate that DNA Damages accumulate in the absence of FAM111A upon niraparib treatment. It would be informative to have a better characterization of the replication stress response induced in response to unresolved PARP-DNA complexes. Which pathway is activated: ATR-Chk1 and/or ATM-Chk2 ? Does ssDNA accumulate at replication forks ?”

- *FAM111A* KO cells showed augmented phospho-Chk1, but not phospho-Chk2, compared to WT cells after PARPi treatment. The results were included in **Fig. 6c** in the revised manuscript. On the other hand, we were not able to detect ssDNA accumulation in either WT or *FAM111A* KO cells after PARPi treatments using BrdU immunostaining in non-denaturing condition [*Methods Mol Biol* 1292: 67-75 (2015)], although it was apparent in HU-treated cells. Given that Chk1 is activated and that activation of ATR-Chk1 pathway is dependent on exposed ssDNA [*Nat Rev Mol Cell Biol* 18: 622-636 (2017)], we speculate that ssDNA accumulation after PARPi treatments might have been below the detection limit of the assay. Nonetheless, our data of Chk1 phosphorylation further support our conclusion that *FAM111A* KO cells experience increased replication stress after PARPi treatment.

Minor points:

“Statistics are missing on Fig 4c, 4f, S2b, S2c, S2d, S2f, S5c”

- Statistics were added to those figures.

“Fig 4c: stars on the legend and triangles on the survival curves. To be clarified.”

- Symbols in the graph were clarified.

In summary, we have endeavored to address all of the comments of the Reviewers. We thank them again for their insightful suggestions.

REVIEWERS' COMMENTS:

Reviewer #1 (Remarks to the Author):

I have now read the revised manuscript and would like to congratulate the authors for a thorough and important work relating to the impact of protein roadblocks to replication. All my comments have been addressed.

Reviewer #3 (Remarks to the Author):

The authors have carried out a fantastic work which reinforces the solidity of their conclusions. All the criticisms were addressed convincingly.